# MambaQuant: Quantizing the Mamba Family with Variance Aligned Rotation Methods

**Zukang Xu**[1*], **Yuxuan Yue**[1,2*†], **Xing Hu**[1], **Zhihang Yuan**[1], **Zixu Jiang**[1,3†],
**Zhixuan Chen**[1], **Jiangyong Yu**[1], **Chen Xu**[1], **Sifan Zhou**[1,4†✉], **Dawei Yang**[1✉]

[1]Houmo AI      [2]Harbin Institute of Technology (Shenzhen)
[3]Nanjing University    [4]Southeast University
zukang.xu@houmo.ai    23s151163@stu.hit.edu.cn
sifanjay@gmail.com     dawei.yang@houmo.ai

## Abstract

Mamba is an efficient sequence model that rivals Transformers and demonstrates significant potential as a foundational architecture for various tasks. Quantization is commonly used in neural networks to reduce model size and computational latency. However, applying quantization to Mamba remains underexplored, and existing quantization methods, which have been effective for CNN and Transformer models, appear inadequate for Mamba models (e.g., Quarot suffers a 21% accuracy drop on Vim-T[†] even under W8A8). We have pioneered the exploration of this issue and identified several key challenges. First, significant outliers are present in gate projections, output projections, and matrix multiplications. Second, Mamba's unique parallel scan further amplifies these outliers, leading to uneven and heavy-tailed data distributions. Third, even with the application of the Hadamard transform, the variance across channels in weights and activations still remains inconsistent. To these ends, we propose MambaQuant, a post-training quantization (PTQ) framework consisting of: 1) Karhunen-Loève Transformation (KLT) enhanced rotation, rendering the rotation matrix adaptable to diverse channel distributions. 2) Smooth-Fused rotation, which equalizes channel variances and can merge additional parameters into model weights. Experiments show that MambaQuant can quantize both weights and activations into 8-bit with less than 1% accuracy loss for Mamba-based vision and language tasks. To the best of our knowledge, MambaQuant is the first comprehensive PTQ design for the Mamba family, paving the way for further advancements in its application.

## 1 Introduction

Mamba (Gu & Dao, 2023) is a modern sequence model that competes with the Transformer (Vaswani et al., 2017), particularly noted for its ability to handle extremely long sequences. The model's design is inspired by the Structured State Space model (S4) (Gu et al., 2021) and integrates features from recurrent, convolutional, and continuous-time models to effectively capture long-term periodic dependencies. Expanding upon the S4 paradigm, Mamba brings about several noteworthy improvements, especially in handling time-variant operations. These enhancements enable the effective and efficient processing of lengthy data sequences, positioning Mamba as a promising foundational architecture for vision (Zhu et al., 2024; Liu et al., 2024), language (Gu & Dao, 2023; Li et al., 2024), and multi-modality tasks (Zhao et al., 2024).

Quantization is an essential technique for deploying deep neural networks (DNNs) in environments with limited computational resources and the demand for real-time processing. This process involves converting weights and activation of neural networks from high precision (e.g., 32-bit floating point numbers) to lower precision (e.g., 8-bit integers) to reduce memory usage, computational burden, and energy consumption. Although quantization has been successfully utilized in convolutional

---

✉Corresponding author
*Equal contribution
†This work was conducted during his internship at Houmo AI

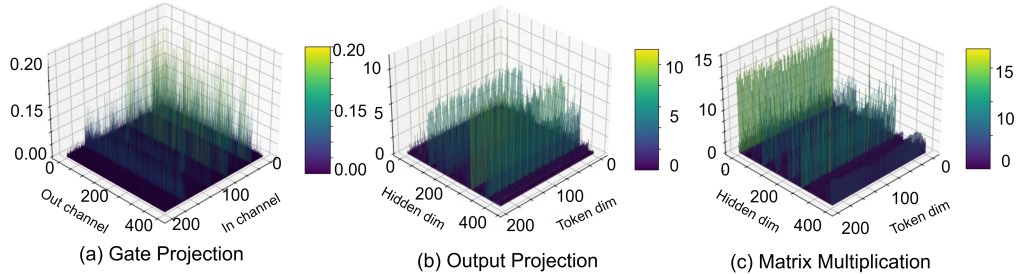

(a) Gate Projection  (b) Output Projection  (c) Matrix Multiplication

Figure 1: Visualized distribution of hard layers for Mamba quantization. (a) denotes the weight of the gate projection, (b) denotes the input activations of the output projection. (c) represents the output of the parallel scan (PScan) operator, which is also one of the input to the matrix multiplication.

neural networks (Krishnamoorthi, 2018; Liu et al., 2023) and Transformer-based large language models (T-LLMs) (Du et al., 2024; Yuan, 2024), its application within the Mamba family has not been systematically analyzed or studied.

To establish a comprehensive quantization methodology for Mamba models, we first examine the potential constraints and challenges involved: ❶ **Significant outliers occur in both weights and activations of Mamba models.** We observe the presence of outliers in the weights of linear layers, particularly in the gate projection layers (Figure 1(a)) of Mamba-LLM (Gu & Dao, 2023) for language tasks. We also find that certain inputs to linear layers exhibit significant variance in the channel dimension. This occurrence is particularly pronounced in the output projection layers (Figure 1(b)) of Vim (Xu et al., 2024) for vision tasks. ❷ **Parallel Scan (PScan) further amplifies the outliers of activations.** To obtain hidden states at each timestamp, the PScan operator (Smith et al., 2022) continuously performs self-multiplication of a fixed parameter matrix. In this case, channels exhibiting higher values will be amplified, while those with comparatively lower values will be diminished. This obvious numerical difference across channels is directly expanded to activations (e.g., input variable to the matrix multiplication as shown in Figure 1(c)).

Given that both Mamba and Transformer are sequence models with fully connected layers to be quantized, our initial solution involves exploring techniques that have been proven effective on Transformer-based large language models (T-LLMs). Recently, Hadamard-based methods (Tseng et al., 2024), known for the capacity to uniform maximum values and the equivalent transformation property, have shown significant success in the quantization of T-LLMs. For instance, quantizing LLAMA2-70B to 4 bits with QuaRot (Ashkboos et al., 2024b) maintains 99% of the zero-shot performance. However, directly applying this method to Mamba models leads to significant accuracy degradation (e.g., on average more than 12% accuracy drop on Vim (Xu et al., 2024) even at 8 bits). Our analysis reveals that Hadamard transformation fails to achieve variance alignment across channels, as shown in Figure 2(b)(e). The inconsistent variances inevitably result in an uneven numerical distribution of the quantization data, thereby decreasing the accuracy.

To this end, we propose *MambaQuant*, an effective and efficient post-training quantization (PTQ) framework tailored for Mamba models. The central concept of MambaQuant is to resolve the issue of inconsistent variances arising from the Hadamard transformation, thereby promoting the Mamba quantization. Specifically, MambaQuant considers two distinct situations depending on whether to integrate the rotation matrix into weights: the offline mode for exclusion and the online mode for inclusion. (1) We propose the Karhunen-Loève Transformation (KLT) enhanced rotation in the offline mode. This technique multiplies the Hadamard matrix with the KLT matrix, enabling the rotation matrix to accommodate various channel distributions. (2) We introduce the smooth-fused rotation in the online mode. This approach performs smoothing before the Hadamard transformation. The additional smoothing parameters are flexibly integrated into weights of Mamba blocks to avoid extra cost of memory space and inference step. Consequently, both the maximum values and the variances of the quantization data are sufficiently normalized in the channel dimension (i.e., they are consistent for the offline mode and closely aligned for the online mode as shown in Figure 2(c)(e)).

Experiments show that MambaQuant outperforms existing methods across various tasks on different Mamba model families, including Vim (Zhu et al., 2024) and Mamba-ND (Li et al., 2024) for Mamba-based vision tasks, as well as Mamba-LLM (Gu & Dao, 2023) for Mamba-based language

tasks. MambaQuant quantizes both weights and activations into 8-bit with a slight accuracy drop (**less than 1%**) for all models. Additionally, it can quantize weights to 4-bit with a minimal accuracy drop (about 1%) for vision tasks, and achieves significant accuracy improvements in language tasks compared to existing methods. Lastly, our contributions can be concluded as follows:

- We identify that Mamba encounters quantization challenges primarily due to significant outliers, which are even amplified by PScan. Our analysis reveals that the Hadamard transformation is hindered by inconsistent channel variances to effectively solve these problems.

- We propose MambaQuant. For offline mode, we introduce the KLT-Enhanced rotation to equalize the channel variances. For online mode, we introduce smooth-fused rotation to normalize the channel variances. Both the offline and online transformation can achieve more uniform distributions prior to the quantization process.

- To the best of our knowledge, MambaQuant is the first comprehensive PTQ framework for the Mamba family. It can efficiently quantize both weights and activations into 8-bit with less than 1% accuracy loss for Mamba-based vision and language tasks.

- As a pioneering study on quantization within the Mamba family, we have published the code in the hope of promoting further research and facilitating advancements in this field.

## 2 RELATED WORK

**Mamba Models**  Mamba (Gu & Dao, 2023) is a selective structured state space model that substantially improves the performance of state space models (SSM) in handling sequential data. It transforms parameters in the structured state space model (S4) (Gu et al., 2021) into learnable functions and proposing a parallel scanning method. By overcoming the local perception limitations of convolutional neural networks (CNNs) (Krizhevsky et al., 2012; Yang et al., 2024) and the quadratic computational complexity of Transformers (Vaswani et al., 2017), Mamba-based networks (Xu et al., 2024) are widely applied in various tasks. For instance, the original Mamba (Gu & Dao, 2023) demonstrates comparable performance to Transformers in language modeling, audio generation, and DNA sequence prediction. Vision Mamba (Vim) (Zhu et al., 2024) marks the first introduction of Mamba to the field of computer vision, employing bidirectional SSM for global modeling and position embedding for position-aware understanding. Subsequently, VMamba (Liu et al., 2024) proposes cross-scan module to address the direction-sensitive challenges. LocalMamba (Huang et al., 2024) further improves performance by incorporating local inductive biases, while PlainMamba (Huang et al., 2024) is designed as a non-hierarchical structure for enhancing integration across the different

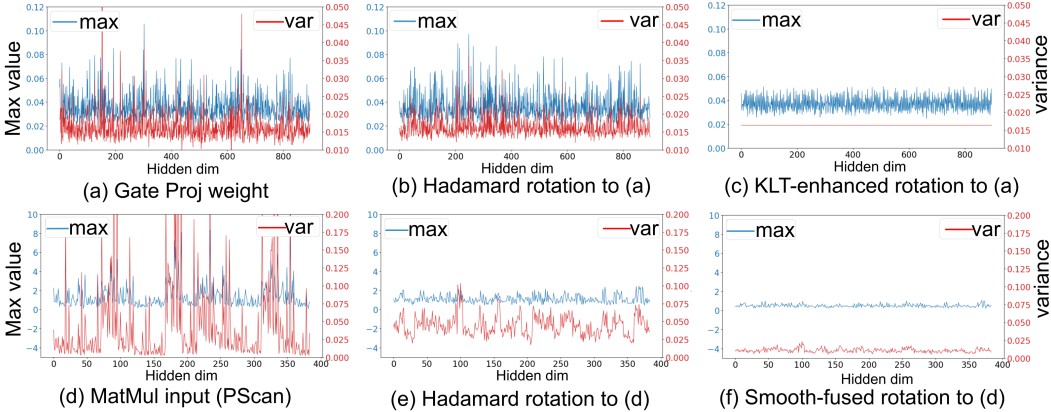

Figure 2: Maximum values (blue color) and variances (red color) distribution across channels of: (a) the original weight of the gate projection; (b) applying the standard offline Hadamard rotation to (a); (c) applying the proposed KLT-Enhanced rotation to (a); (d) the input activation (generated by PScan) of the matrix multiplication; (e) applying the standard online Hadamard rotation to (d); (f) applying the proposed smooth-fused rotation to (d).

scales. Mamba-ND (Li et al., 2024) simply alternates the order of sequence, effectively extending Mamba to multi-modal data including images and videos. Despite reduced computational demands and impressive performance, the large size of these models still limits their application on edge devices.

**Quantization Methods.** Quantization is an effective model compression technique. Current methods can be categorized into quantization aware training (QAT) and post training quantization (PTQ). While QAT typically necessitates full parameters training, which poses challenges for large models, PTQ (Zhou et al., 2024; Yue et al., 2024) has garnered more research attention. Quantizing full-precision variables of pre-trained models into low-bit integers, PTQ reduces the memory consumption and enhances the inference speed. For instance, in the field of Vision Transformer (Dosovitskiy, 2020), FQ-ViT (Lin et al., 2021) introduces a comprehensive quantization scheme for the first time, employing powers of two factors and Log2 quantizers for layer normalization and attention mapping. RepQ-ViT (Li et al., 2023) further addresses the issue of extreme distribution in activations after layer normalization and SoftMax operations. In the field of Large Language Models (LLMs), GPTQ (Frantar et al., 2022) introduces a layer-wise quantization technique based on approximate second-order information, quantizing weights to 3-4 bit with minimal accuracy loss. To suppress outliers in activations, SmoothQuant (Xiao et al., 2022) adopts a smoothing parameter that transfers the difficulty of quantizing activations to weights. Recently, QuaRot (Ashkboos et al., 2024b) adopts a similar methodology, which combines the rotation in QuIP (Chee et al., 2024) and the computational invariance in SliceGPT (Ashkboos et al., 2024a), pushing PTQ to a new level. While these methods perform effectively for Transformer-based large language models, they do not work well with mamba models. Notably, to our knowledge, our method is the first PTQ solution specifically designed for Mamba models, applicable to both Mamba-based vision and language tasks.

## 3 PRELIMINARIES

### 3.1 STATE SPACE MODELS

The state space models (SSMs) are typically regarded as contiguous linear time-invariant (LTI) systems (Kalman, 1960), which map an input signal $x(t) \in \mathbb{R}$ to its output $y(t) \in \mathbb{R}$ through a hidden state $h(t) \in \mathbb{R}^{d \times 1}$:

$$h(t) = Ah(t-1) + Bx(t), \qquad (1)$$

$$y_{ssm}(t) = Ch(t) + Dx(t), \qquad (2)$$

where $A \in \mathbb{R}^{d \times d}$, $B \in \mathbb{R}^{d \times 1}$, $C \in \mathbb{R}^{1 \times d}$, $D \in \mathbb{R}^{1 \times 1}$ are weighting parameters, $t \in \mathbb{Z}^+$, and $h(0)$ is an initial hidden state.

### 3.2 MAMBA ARCHITECTURE

Since the usage of LTI system, the model parameters remain unchanged, decreasing the performance when representing changing inputs. To tackle this

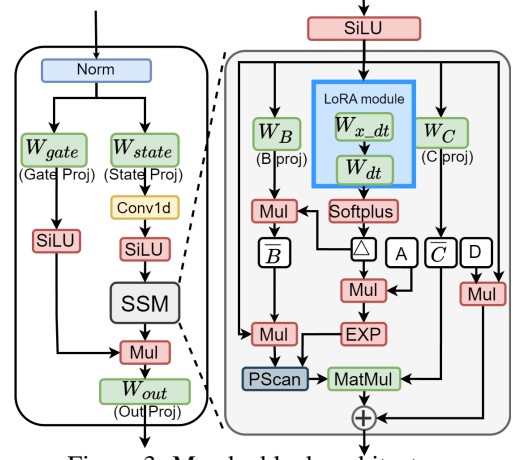

Figure 3: Mamba block architecture.

issue, Mamba (Gu & Dao, 2023) propose an implementation of selective SSM (Gu et al., 2021), which formulates parts of the parameters as functions of a specific input sequence:

$$x' = \sigma(\text{DWConv}(\text{State\_Projection}(x))), \quad \Delta = \text{Softplus}(\text{LoRA\_Module}(x')), \qquad (3)$$

$$\overline{A} = e^{A \odot \Delta}, \quad \overline{B} = \text{B\_Projection}(x') \odot \Delta, \quad \overline{C} = \text{C\_Projection}(x'), \qquad (4)$$

where $x'$ denotes the transformed input and $\sigma$ represents the SiLU activation. Those input-dependent parameters and $x'$ are used by the Parallel Scan (PScan) operator to generate $y'_{ssm}$, The calculation process of PScan can be expressed as:

$$h(t) = \overline{A}h(t-1) + \overline{B}x(t), \quad y'_{ssm}(t) = \overline{C}h(t), \qquad (5)$$

This temporary output is then element-wisely multiplied with a gated variable $z$ to generate better outputs:

$$z = \sigma(\text{Gate\_Projection}(x)), \quad y_{out} = y'_{ssm} \odot z. \qquad (6)$$

### 3.3 QUANTIZATION

Quantization is generally performed to obtain a low-precision representation (e.g., 4-bit integer) from a high-precision variable (e.g., 16-bit floating points). For a tensor $\boldsymbol{x}$ to be quantized, it can be uniformly quantized to $b$-bits as follows (Jacob et al., 2018):

$$\hat{\boldsymbol{x}} = (\text{clamp}(\lfloor \frac{\boldsymbol{x}}{\boldsymbol{s}} \rceil + \boldsymbol{z}, 0, 2^b - 1) - \boldsymbol{z}) \cdot \boldsymbol{s}, \quad \boldsymbol{s} = \frac{\max(\boldsymbol{x}) - \min(\boldsymbol{x})}{2^b - 1}, \quad \boldsymbol{z} = \frac{-\min(\boldsymbol{x})}{\boldsymbol{s}}, \quad (7)$$

where $\boldsymbol{z}$ is the zero point, $s$ is the scale factor, $\lfloor \cdot \rceil$ denotes the rounding-to-nearest operator, $\text{clamp}$ is the clipping function.

## 4 METHOD

### 4.1 DIMINISHED EFFECTIVENESS OF HADAMARD TRANSFORMATION

Hadamard transformation is a promising quantization method for LLMs, recognized for its effectiveness in handling outliers and its computational simplicity and speed. It provides robust performance while efficiently managing data variability.

Hadamard matrices are square matrices with orthogonal rows and columns, where each element is either $\frac{1}{\sqrt{m}}$ or $-\frac{1}{\sqrt{m}}$ ($m$ is the order of the Hadamard matrix). By multiplying with such a uniformly distributed matrix, each row contributes relatively equally to a given channel, thereby making the extreme values of the channels closer together (Tseng et al., 2024). Additionally, due to the orthogonal nature, the Hadamard matrix can be well integrated into model weights while ensuring the computational consistency.

We initially attempt to directly apply this method to Mamba models, particularly to the gate projection, output projection, and the matmul layer. However, the Hadamard transformation is not sufficiently effective in normalizing the hard layers mentioned in Figure 1 of the Mamba architecture with significant outliers, as illustrated in Figure 2(b)(e).

To this end, we conduct a thorough analysis of this issue and find that this method fails to align the channel variance of quantization variables, thereby overlooking the distribution consistency between different channels. In detail, given a centered data matrix (the columns of the matrix are zero-mean.) $\boldsymbol{X}$ (weights or activations) with dimensions $(n, m)$ and the Hadamard transformation matrix $\boldsymbol{H}$ with dimensions $(m, m)$, the covariance matrix $\boldsymbol{C}_{XH}$ of the transformed matrix $\boldsymbol{X}\boldsymbol{H}$ can be expressed as:

$$\boldsymbol{C}_{XH} = \frac{1}{n-1}(\boldsymbol{X}\boldsymbol{H})^T \boldsymbol{X}\boldsymbol{H} = \frac{1}{n-1}\boldsymbol{H}^T \boldsymbol{X}^T \boldsymbol{X}\boldsymbol{H} = \frac{1}{n-1}\boldsymbol{H}^T \boldsymbol{K}\boldsymbol{\Lambda}\boldsymbol{K}^T \boldsymbol{H}, \quad (8)$$

where $\boldsymbol{X}^T \boldsymbol{X} = \boldsymbol{K}\boldsymbol{\Lambda}\boldsymbol{K}^T$ represents the eigenvalue decomposition, $\boldsymbol{K}$ is the eigenvectors matrix, and $\boldsymbol{\Lambda}$ is the diagonal eigenvalues matrix. Considering that $\boldsymbol{H}^T \boldsymbol{K}$ and $\boldsymbol{K}^T \boldsymbol{H}$ are transposed matrices of each other, the $l$-th diagonal elements of $\boldsymbol{C}_{XH}$ can be expressed as:

$$(\boldsymbol{C}_{XH})_{ll} = \frac{1}{n-1}\sum_{j=1}^{m}(\boldsymbol{H}^T \boldsymbol{K})_{lj}^2 \lambda_j = \frac{1}{n-1}\sum_{j=1}^{m}(\sum_{i=1}^{m}\boldsymbol{H}_{il}\boldsymbol{K}_{ij})^2 \lambda_j, \quad (9)$$

where $\lambda_j$ is the $j$-th eigenvalue of $\boldsymbol{\Lambda}$. The complete derivation from Equation 8 to Equation 9 is provided in Appendix A.2. For a given value of $l$, Equation 9 represents the variance of the $l$-th channel. As the vector $\boldsymbol{H}_{:,j}$ varies with the $l$, the channel variances cannot be proven to be numerically close in most cases refers to Appendix A.3. Further, considering that $\boldsymbol{H}$ is a fixed matrix while both $\boldsymbol{K}$ and $\lambda$ are input-dependent, it is not feasible for the Hadamard transformation to uniformly adjust the channel variances across all scenarios. This property of Hadamard transformation inevitably formulates a distinct distribution for each channel, thus leading to sub-optimal quantization.

### 4.2 KLT-ENHANCED ROTATION FOR OFFLINE TRANSFORMATION

To overcome the constrain stated in Section 4.1, we introduce the Karhunen-Loève Transformation (KLT) (Dony et al., 2001) to equalize channel variances. KLT identifies principal components in the data and projects it onto these components, retaining the most critical information by focusing on directions of maximum variance. In practical, the mean value for each channel of the Mamba weights

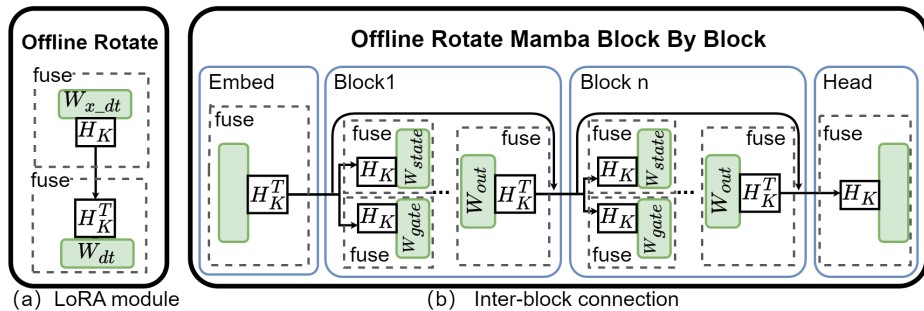

Figure 4: Offline transformation designs utilizing the KLT-Enhanced rotation.

and activations is typically close to zero, meeting the applicable conditions of KLT. Specifically, We apply KLT by performing eigenvalue decomposition on the covariance matrix $C_X$ of the centered matrix $X$ derived from the calibration data.

$$C_X = \frac{1}{n-1}X^T X = \frac{1}{n-1}K\Lambda K^T. \tag{10}$$

Next, the KLT-Enhanced rotation matrix $H_K$ can be obtained by applying the KLT to the Hadamard matrix $H$, as described in Equation 11, and the Equation 8 turns into Equation 12:

$$H_K = KH, \tag{11}$$

$$C_{XH_K} = \frac{1}{n-1}H_K^T K\Lambda K^T H_K = \frac{1}{n-1}H^T K^T K\Lambda K^T KH = \frac{1}{n-1}H^T I\Lambda IH, \tag{12}$$

where $I$ denotes the identity matrix. Consequently, the Equation 9 thus turns to Equation 13:

$$(C_{XH_K})_{ll} = \frac{1}{n-1}\sum_{j=1}^{m}(\sum_{i=1}^{m}H_{il}I_{ij})^2\lambda_j = \frac{1}{(n-1)m}\sum_{j=1}^{m}\lambda_j. \tag{13}$$

In this way, the variance of each channel becomes the same, making quantization much easier. This transformation serves a dual purpose: it not only equalizes the variance among different channels but also embodies the distinctive property of Hadamard matrices, which is their ability to balance maximum values. We also provide detailed steps for the formula of performing KLT rotation followed by Hadamard rotation in Appendix A.4 to achieve variance balancing. In practice the KLT is offline performed by using the calibration data to avoid extra computational costs. Still it can be well-generalized to wider range of inputs (detailed in Appendix A.7).

To apply this KLT-Enhanced rotation matrix, we modify the offline transformation in QuaRot (Ashkboos et al., 2024b) for the Mamba structure. As shown in Figure 4, we employ this strategy for the LoRA module and the inter-block connection (where the output projection, gate projection and the state projection is transformed).

## 4.3 SMOOTH-FUSED ROTATION FOR ONLINE TRANSFORMATION

To mitigate the shortcoming of the Hadamard rotation discussed in Section 4.1 where the online transformation is applied, we introduce the smoothing technique prior to its execution. The motivation of employing this method is to uniform the channel variances through a smoothing vector. Typically, the smoothing factors can be absorbed into the neighbored layers with the quantization of T-LLMs (Xiao et al., 2022; Shao et al., 2023). This operation effectively circumvents the demand for additional memory allocation and computational overhead that would arise from the incorporation of extra parameters. However, this approach does not align with the Mamba modules due to the non-linear SiLU operation and the complex loop structure of PScan. To this end, two distinct designs are proposed for output projection and matrix multiplication, respectively.

**For the output projection layer:** We improve the traditional SiLU activation function with Smooth SiLU (S-SiLU) (Hu et al., 2024) to meet the needs of smooth-fused quantization:

$$\text{S-SiLU}(x, s) = x \odot \sigma(s \odot x), \tag{14}$$

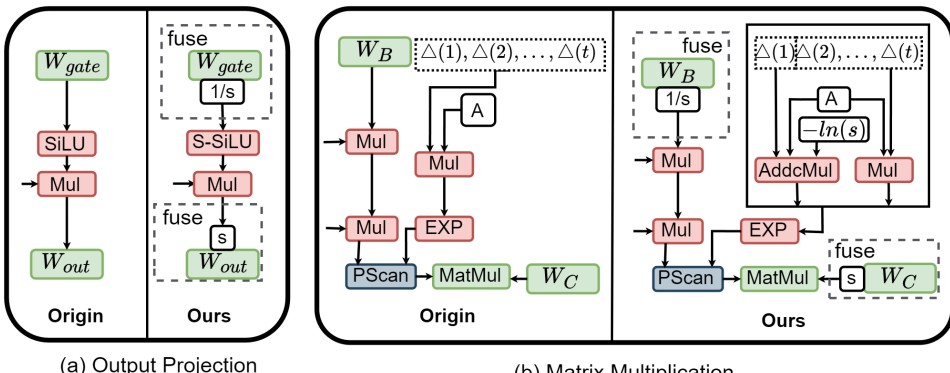

(a) Output Projection     (b) Matrix Multiplication

Figure 5: Fusing smooth parameters into the Mamba structure.

where $\boldsymbol{x}$ is an activation variable, $\sigma(\cdot)$ represents the Sigmoid function, $\boldsymbol{s}$ denotes the introduced smoothing parameter, and '$\odot$'represents element-wise multiplication. Depicted in Figure 5(a), the application of the S-SiLU function on the gate projection described by Equation 6 can be expressed as follows:

$$\boldsymbol{y}_{out} = [\boldsymbol{y}_{ssm} \odot \text{SiLU}(\boldsymbol{x}_g \boldsymbol{W}_g)]\boldsymbol{W}_o = [\boldsymbol{y}_{ssm} \odot \text{S-SiLU}(\boldsymbol{x}_g \boldsymbol{W}_g', s_{out})]\boldsymbol{W}_o', \tag{15}$$

where $\boldsymbol{y}_{ssm}$ denotes the output activation of the SSM, $\boldsymbol{W}_g' = \boldsymbol{W}_g \oslash \boldsymbol{s}_{out}$ and $\boldsymbol{W}_o' = \boldsymbol{s}_{out} \odot \boldsymbol{W}_o$ are transformed weights of the gate projection (denoted with subscript 'g') and the output projection (denoted with subscript 'o'), '$\oslash$'represents element-wise division, $\boldsymbol{s}_{out}$ is the absorbed smoothing factor, $\boldsymbol{x}_g$ is the input of the gate projection, and $\boldsymbol{y}_{out}$ represents the final output of the Mamba block.

**For the matrix multiplication layer:** We also design a scheme to absorb the smoothing factor for the matrix multiplication operator within the Mamba block. One input stream of the multiplication is the output of the $\boldsymbol{C}$ projection in Equation 4, which can directly fuse the smoothing factor $s_{mm}$ into the weight of C projection ($W_C$) as shown in Figure 5(b). Another input stream comes from the output of the parallel scan operator. As shown in Equation 5, the calculation of PScan includes addition operator, and the smoothing factor $s_{mm}$ will be transmitted along two routes on both sides of the addition operator. One route is transmitted through $\overline{\boldsymbol{B}}$ and absorbed by the weight of $\boldsymbol{B}$ projection ($W_B$), and the other route is transmitted through $\overline{\boldsymbol{A}}$ and absorbed by $\Delta$, which defined in Equation 3. Because of the existence of exponential calculation in Equation 4, $1/s_{mm}$ becomes $-ln(s_{mm})$ when transmitted to $\Delta$, and is absorbed by applying the addcmul operator (PyTorch, 2023) to $\Delta(1)$ in Euation 16. It is solely applied to the first token of $\Delta$ ($\Delta(1)$).

$$\text{addcmul}(-\ln(\boldsymbol{s}_{mm}), \boldsymbol{\Delta}(1), \boldsymbol{A}) = \boldsymbol{A}\boldsymbol{\Delta}(1) - \ln(\boldsymbol{s}_{mm}). \tag{16}$$

After smoothing, the channel variances of activations for the output projection and the matrix multiplication becomes relatively uniform. Subsequently, we modify and apply the online Hadamard rotation (Ashkboos et al., 2024b) for the Mamba structure as shown in Figure 6. The Hadamard matrix $\boldsymbol{H}$ is dynamically applied to the input activation of the output projection and the matrix multiplication, while the transposed $\boldsymbol{H}^T$ can be absorbed into corresponding weights.

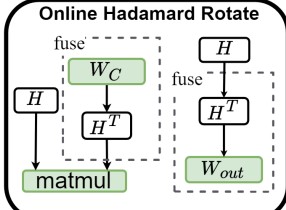

Figure 6: Online transformation designs utilizing the smooth-fused rotation.

## 5 EXPERIMENTS

**Models and datasets.** We assess the general quantization capabilities of our proposed MambaQuant framework across three representative Mamba-based applications: Mamba (Gu & Dao, 2023), Vim (Zhu et al., 2024), and Mamba-ND (Li et al., 2024). We evaluate the performance of the quantized Mamba model across vision and language tasks. For vision tasks, we tested the model on the image classification dataset ImageNet (Russakovsky et al., 2015) and the video classification dataset UCF-101 (Soomro et al., 2012). In the language domain, we conducted evaluations on five standard datasets: ARC-E (Boratko et al., 2018),

Table 1: Comparative results under different quantization settings for Vision Mamba models. The Vim models and Mamba-2d models are tested for accuracy on ImageNet, while the Mamba-3d model is tested for accuracy on UCF-101. $^\dagger$ indicates the fine-tuned model on Vim. $^\ddagger$ denotes results based on official weights.

| Bit Width | Methods | Vision Mamba | | | | | Mamba-ND | | |
|---|---|---|---|---|---|---|---|---|---|
| | | Vim-T | Vim-T$^\dagger$ | Vim-S | Vim-S$^\dagger$ | Vim-B | mamba-2d S | Mamba-2d B | Mamba-3d |
| FP16 | - | 76.1 | 78.3 | 80.5 | 81.6 | 80.3$^\ddagger$ | 81.7 | 83.0 | 89.6 |
| W8A8 | RTN | 37.4 | 32.4 | 68.8 | 68.8 | 52.2 | 80.3 | 82.2 | 87.9 |
| | GPTQ+RTN | 37.7 | 32.5 | 68.9 | 70.5 | 52.2 | 80.4 | 82.2 | 87.8 |
| | SmoothQuant | 37.7 | 32.3 | 68.7 | 72.9 | 52.1 | 80.3 | 82.2 | 87.9 |
| | QuaRot | 59.3 | 57.4 | 73.8 | 75.5 | 73.8 | 80.8 | 82.3 | 88.0 |
| | Ours | 75.6 | 77.8 | 80.3 | 81.4 | 80.1 | 81.2 | 82.8 | 89.0 |
| W4A8 | RTN | 26.3 | 25.0 | 66.1 | 70.0 | 46.2 | 40.6 | 78.8 | 86.1 |
| | GPTQ+RTN | 30.4 | 27.9 | 66.5 | 70.6 | 47.7 | 60.3 | 78.9 | 86.8 |
| | SmoothQuant | 27.0 | 26.0 | 66.4 | 70.2 | 46.7 | 59.7 | 80.2 | 86.9 |
| | QuaRot | 52.7 | 48.5 | 72 | 74.0 | 72.8 | 80.1 | 82.0 | 86.9 |
| | Ours | 72.1 | 73.7 | 79.4 | 80.4 | 79.8 | 80.4 | 81.9 | 88.4 |

ARC-C  (Clark et al., 2018), PIQA  (Bisk et al., 2020), Winogrande  (Sakaguchi et al., 2021), and HellaSwag  (Zellers et al., 2019), and reported the average performance across these datasets.The metric used for the evaluation of our test results on these tasks is Accuracy (Acc).

**Baselines and implementation details.**    For comparison, we apply different quantization settings to the Mamba model and reported the performance under two configurations: W8A8 and W4A8 (weights and activations). Additionally, we compare with the different quantization methods, including the Round To Nearest (RTN) method, SmoothQuant (Xiao et al., 2022), GPTQ  (Frantar et al., 2022) for weights and RTN for activations (GPTQ+RTN), as well as QuaRot (Ashkboos et al., 2024b). For the vision tasks, we utilize a static quantization approach. The calibration data for image classification was randomly sampled from 128 images in the ImageNet  (Russakovsky et al., 2015) test set, while for video classification, we used samples from the UCF-101  (Soomro et al., 2012) test set for calibration. In contrast, for language tasks, we employ dynamic quantization to better adapt to the varying input structures during inference.

## 5.1   OVERALL RESULTS

**Performance Comparison on Vision Model.**    Table 1 presents the results of various Mamba vision models under different quantization settings, including Vision Mamba and Mamba-ND. The quantization configurations evaluated are W8A8 (8-bit weights and activations) and W4A8 (4-bit weights and 8-bit activations). The table compares the performance of several quantization methods, including RTN, GPTQ+RTN, SmoothQuant, QuaRot, and our proposed method ("Ours"), across different Mamba model variants. Our proposed quantization method demonstrates a significant improvement over baseline techniques. Under the W8A8 configuration, the performance of our method remains within 1 points of the floating-point baseline accuracy. In the stricter W4A8 setting, our method substantially outperforms competing approaches, which experience more pronounced accuracy drops. These results indicate that our approach offers a more robust solution for maintaining high accuracy in Vim and Mamba-ND models under quantized settings. The findings suggest that our method is more resilient to the challenges of precision reduction and provides a both practical and effective quantization solution for deploying Mamba models.

**Performance Comparison on Language Model.**    Table 2 illustrates the quantization results for the Mamba model applied to language tasks. The models evaluated range from Mamba-370m to Mamba-2.8b. The table compares the accuracy of several quantization methods, includingRTN, GPTQ+RTN, SmoothQuant, QuaRot, and our method. Our approach delivers superior performance across different model sizes, particularly under the more challenging W4A8 configuration, where it consistently outperforms baseline techniques by a significant margin. These results demonstrate the robustness and efficiency of our quantization method for language model tasks.All experimental results for both vision and language models under our proposed method are provided in Appendix A.6.

Table 2: Comparative results under different quantization settings on language mamba models. Evaluations on the five standard datasets—ARC-E, ARC-C, PIQA, Winogrande and HellaSwag—resulted in the reported average accuracy across these datasets.

| Bit Width | Methods | Mamba-LLM | | | |
|---|---|---|---|---|---|
| | | Mamba-370m | Mamba-790m | Mamba-1.4b | Mamba-2.8b |
| FP16 | - | 50.9 | 54.8 | 58.6 | 62.2 |
| W8A8 | RTN | 45.7 | 44.9 | 53.9 | 58.4 |
| | GPTQ+RTN | 46.2 | 48.6 | 55.0 | 58.9 |
| | SmoothQuant | 45.2 | 41.7 | 54.2 | 58.7 |
| | QuaRot | 48.8 | 51.6 | 56.9 | 59.3 |
| | Ours | 50.0 | 53.8 | 58.3 | 62.1 |
| W4A8 | RTN | 36.2 | 35.4 | 51.6 | 54.8 |
| | GPTQ+RTN | 36.7 | 36.0 | 51.1 | 53.6 |
| | SmoothQuant | 36.8 | 39.3 | 52.0 | 54.9 |
| | QuaRot | 43.4 | 40.0 | 53.8 | 58.5 |
| | Ours | 43.9 | 45.8 | 54.3 | 58.5 |

## 5.2 ABLATION STUDY

Table 3: Ablation Experiment For KLT-Enhanced Rotation.

| Bit Width | Methods | Vim T[†] | Mamba-790m | Bit Width | Methods | Vim T[†] | Mamba-790m |
|---|---|---|---|---|---|---|---|
| FP16 | - | 78.3 | 54.8 | FP16 | - | 78.3 | 54.8 |
| W8A8 | Baseline(RTN) | 32.4 | 44.2 | W4A8 | Baseline(RTN) | 25.0 | 35.4 |
| | Hadamard Rotate | 33.9(↑ 1.5) | 50.8(↑ 6.6) | | Hadamard Rotate | 25.1(↑ 0.1) | 40.2(↑ 4.8) |
| | KLT-Enhanced Rotate | 47.7(↑ 15.3) | 51.3(↑ 7.1) | | KLT-Enhanced Rotate | 38.9(↑ 3.9) | 42.3(↑ 6.9) |

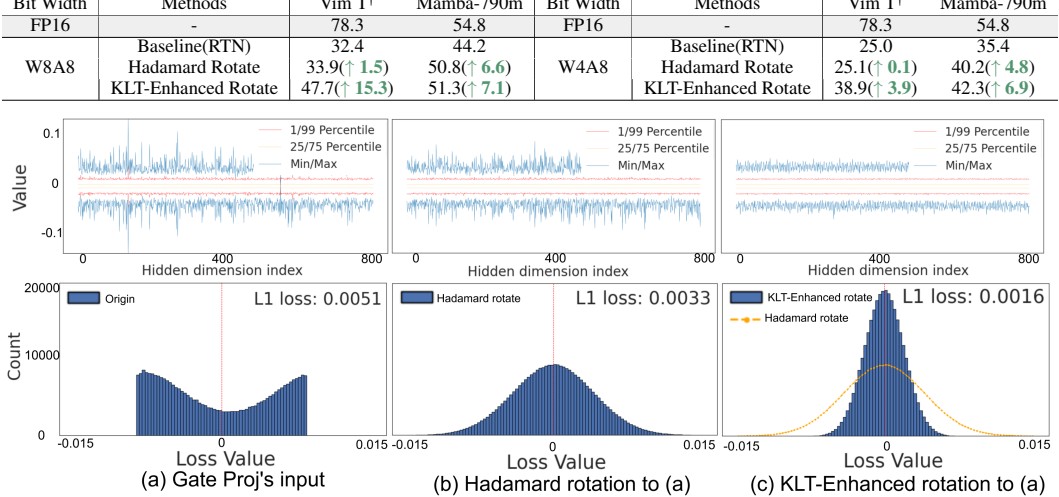

Figure 7: Data of activatin of gate projection distribution and quantization of losses. (a) Original data ; (b) Hadamard rotateion to (a) ; (c) KLT-Enhanced rotation to (a). The first row of graphs depicts the lineshowing the distribution of data values at different quantile points across various channels, while the second row illustrates the count bar graphs representing the different quantization losses.

**Ablation Study on KLT-Enhanced Rotation.**   We conducted a series of ablation studies on the offline rotation layers described in Section 4.2, comparing the KLT-Enhanced rotation and Hadamard rotation techniques as proposed in our method. The ablation study comparisons from Table 3 demonstrate the effectiveness of the KLT-enhanced rotation approach. Compared with the direct use of hadamard rotation, our method can achieve a greater degree of progress improvement, which is about 14% higher than the direct hadamard method in the Vim-T[†] quantization of W8A8.

In addition to the quantitative analysis of the ablation study results, Figure 7 visually conveys the optimization brought by the KLT-enhanced rotation over the Hadamard rotation. The upper part of Figure 7 shows the quantile distribution of data across different channels, while the lower part illustrates the quantization loss distribution using the 4-bit per-tensor method. In the bar graph, the horizontal axis represents quantization loss magnitude, and the vertical axis indicates the data point count. Subfigures (a), (b), and (c) display the quantization losses for the original data, Hadamard-rotated data, and KLT-enhanced rotation data, respectively. Comparing these, the KLT-enhanced rotation clearly outperforms Hadamard rotation in smoothing quantization and reducing losses across channels. Moreover, our method reduces the L1 loss from quantization by nearly half compared to using Hadamard rotation alone.

Table 4: Ablation Experiment For Smoothed Rotation.

| Bit Width | Methods | Vim-T[†] | Mamba-790M | Bit Width | Methods | Vim-T[†] | Mamba-790M |
|---|---|---|---|---|---|---|---|
| FP16 | - | 78.3 | 54.6 | FP16 | - | 78.3 | 58.6 |
| W8A8 | Baseline(KLT-enhanced Rotation) | 47.7 | 51.3 | W4A8 | Baseline(KLT-enhanced Rotation) | 38.9 | 42.3 |
| | Hadamard Rotation | 69.7(↑ 22.0) | 51.8(↑ 0.5) | | Hadamard Rotation | 62.0(↑ 23.1) | 43.0(↑ 0.7) |
| | Smooth-Fused Rotation | 77.8(↑ 30.1) | 53.3(↑ 2.0) | | Smooth-Fused Rotation | 73.7(↑ 34.8) | 45.8(↑ 3.5) |

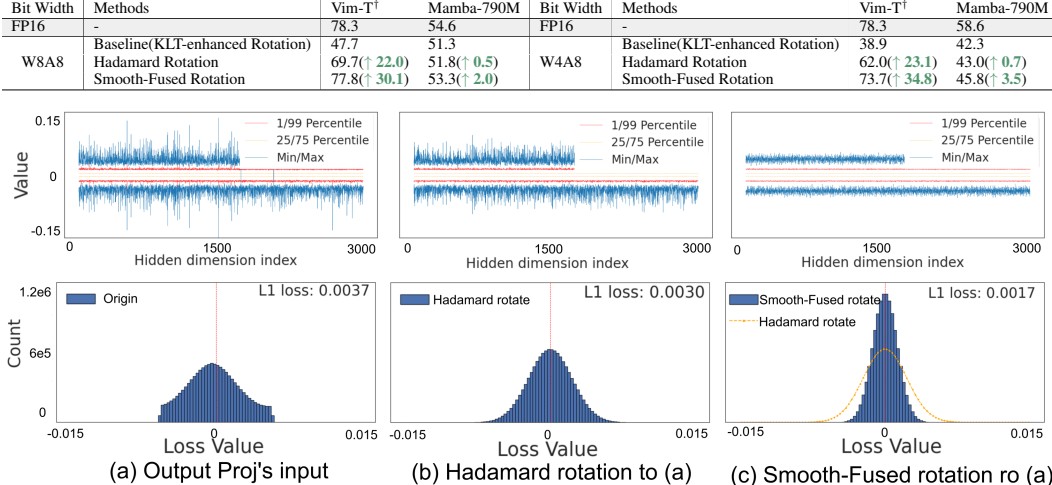

Figure 8: Data of weight of output projection distribution and quantization of losses. (a) Original data quantile and fake quantization loss, (b) Hadamard rotated data quantile and fake quantization loss, (c) Smooth based rotated data quantile and fake quantization loss.

**Ablation Study on Smooth-fused Rotation.** We applied the method described in Section 4.3, focusing on the sensitive matmul and output projection layers. The ablation results comparing the use of online Hadamard rotation and smooth-fused rotation are presented in Table 4. We found that the smooth-fused rotation can lead to significant improvements in quantization accuracy compared to directly using Hadamard rotation. Under the condition of W4A8 quantization configuration, in the Vim T[†] model, the Smooth-Fused rotation method is 11.7% ahead of the direct online Hadamard rotation method in terms of accuracy.

**Memory Occupancy and Computational Cost.** In method 4.3, we introduce a smooth scale parameter to optimize the quantization process, with minimal storage overhead since each quantization channel is represented by a single scalar, which has negligible impact on overall model size. Additionally, the method introduces the application of the online Hadamard rotation technique described in the QuaRot (Ashkboos et al., 2024b), which ensures rapid transformations akin to FFT, minimizing computational impact on inference speed. Taking the Mamba-2.8B model as an example, the smooth scale adds only 329k parameters to the 2.8B model, while for a token sequence length of 1024, the computational increase is 25.6 GFLOPs over the baseline 2.8 TFLOPs. This translates to a 0.01% increase in the parameter size and just a 0.91% increase in computational cost.

## 6 CONCLUSION

In this paper, we focus on introducing the quantization techniques into the realm of Mamba models. Firstly, we identify that significant outliers which challenge the quantization process are present in gate projection, output projection, and matrix multiplication, while the unique PScan operator further amplifies the numerical differences. Secondly, we find that Hadamard transformation method widely adopted in Transformer quantization performs less satisfying when quantizing these hard layers. Our analysis reveals that this method falls short of sufficiently aligning the channel variances, thus remaining an uneven distribution challenging the quantization process. To beyond this limitation, we propose MambaQuant, a comprehensive post training quantization framework especially designed for the Mamba models. The core idea of this strategy is to facilitating the Hadamard transformation with the ability to uniform the variance of each channel, thus enhancing the performance of Mamba quantization. Specifically, we introduce the Karhunen-Loève Transformation to render the rotation matrix adaptable to diverse channel distributions. We also incorporate a smoothing methodology to uniform the channel variances, while additional parameters are fused into model weights to avoid extra overhead. Our proposed MambaQuant advances in accuracy for both Mamba-based vision and language tasks compared to existing methods, making Mamba models more practical for deployment in resource-constrained environments.

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

## A  APPENDIX

### A.1  LIMITATIONS

Recently, Mamba models demonstrates superior accuracy across various vision and language tasks, indicating its strong capabilities in feature extraction and pattern recognition. However, its deployment with the quantization methodology still remains largely under explored. We thus propose MambaQuant, an accurate and efficient post training quantization framework especially desined for the Mamba family.

While this approach can effectively quantize the weights and activations of Mamba models into 8-bit with less than a 1% drop in accuracy, it struggles to maintain such a high level of accuracy when quantizing weights to 4-bit. In addition, we note that the proposed Karhunen-Loève Transformation (KLT) enhanced rotation is efficiently constrained if applied to the online Hadamard rotation. This is primarily due to the additional computation steps introduced by the eigenvalue decomposition (as stated in Equation 10) and the application to the Hadamard matrix (as stated in Equation 11). Despite the constrains, we hope that our work could inspire the research interest on Mamba quantization within the community. We are also committed to extending the KLT-Enhanced rotation method to online transformation in order to achieve better performance in low-bit quantization.

### A.2  DERIVED THE INABILITY OF HADAMARD ROTATION TO ENSURE CONSISTENCY OF COLUMN VARIANCE.

**Hadamard properties.**  An orthogonal matrix $Q$ is a square matrix such that $QQ^T = I$. In this work, we consider only real orthogonal matrices. A rotation matrix is an orthogonal matrix with $|Q| = 1$. A Hadamard matrix is an orthogonal matrix with each element is either $\frac{1}{\sqrt{m}}$ or $-\frac{1}{\sqrt{m}}$. A 2x2 Hadamard matrix is defined as follows:

$$H = \frac{1}{\sqrt{2}} \begin{pmatrix} 1 & 1 \\ 1 & -1 \end{pmatrix} \tag{17}$$

**Covariance Calculation.**  Given a matrix X with dimensions $(n, m)$ with zero mean across each column. Alculate the covariance matrix $C_X$ of X:

$$C_X = \frac{1}{n-1} X^T X = \frac{1}{n-1} K \Lambda K^T, \tag{18}$$

where $K$ is the eigenvectors matrix, $\Lambda$ is the diagonal eigenvalues matrix, $n$ denotes the number of rows. We provide a proof based on the above properties of Hadamard that Hadamard cannot achieve

column variance consistency. In detail, given a Hadamard transformation matrix $\boldsymbol{H}$ with dimensions $(m, m)$.the covariance matrix $\boldsymbol{C}_{XH}$ of the transformed matrix $\boldsymbol{XH}$ can be expressed as:

$$C_{XH} = \frac{1}{n-1}(\boldsymbol{XH})^T \boldsymbol{XH} = \frac{1}{n-1}\boldsymbol{H}^T\boldsymbol{X}^T\boldsymbol{XH}, \tag{19}$$

Subsituate Equation 18 into Equation 19:

$$C_{XH} = \frac{1}{n-1}\boldsymbol{H}^T\boldsymbol{X}^T\boldsymbol{XH} = \frac{1}{n-1}\boldsymbol{H}^T\boldsymbol{K}\boldsymbol{\Lambda}\boldsymbol{K}^T\boldsymbol{H}, \tag{20}$$

Hadamard matrix expansion:

$$\boldsymbol{H} = \begin{pmatrix} H_{11} & H_{12} & \cdots & H_{1m} \\ H_{21} & H_{22} & \cdots & H_{2m} \\ \vdots & \vdots & \ddots & \vdots \\ H_{m1} & H_{m2} & \cdots & H_{mm} \end{pmatrix}, \tag{21}$$

KLT matrix expansion:

$$\boldsymbol{K} = \begin{pmatrix} K_{11} & K_{12} & \cdots & K_{1m} \\ K_{21} & K_{22} & \cdots & K_{2m} \\ \vdots & \vdots & \ddots & \vdots \\ K_{m1} & K_{m2} & \cdots & K_{mm} \end{pmatrix}, \tag{22}$$

We define the matrix $P = H^T K$:

$$\boldsymbol{P} = H^T K = \begin{pmatrix} \sum_{i=1}^m H_{i1}K_{i1} & \sum_{i=1}^m H_{i1}K_{i2} & \cdots & \sum_{i=1}^m H_{i1}K_{im} \\ \sum_{i=1}^m H_{i2}K_{i1} & \sum_{i=1}^m H_{i2}K_{i2} & \cdots & \sum_{i=1}^m H_{i2}K_{im} \\ \vdots & \vdots & \ddots & \vdots \\ \sum_{i=1}^m H_{im}K_{i1} & \sum_{i=1}^m H_{im}K_{i2} & \cdots & \sum_{i=1}^m H_{im}K_{im} \end{pmatrix}, \tag{23}$$

Substitute Equation 23 into Equation 20:

$$\begin{aligned} C_{XH} &= \frac{1}{n-1}\boldsymbol{P}\boldsymbol{\Lambda}\boldsymbol{P}^T \\ &= \frac{1}{n-1}\begin{pmatrix} \sum_{i=1}^m P_{1i}^2\lambda_i & \cdots & \cdots & \cdots \\ \cdots & \sum_{i=1}^m P_{2i}^2\lambda_i & \cdots & \cdots \\ \vdots & \vdots & \ddots & \vdots \\ \cdots & \cdots & \cdots & \sum_{i=1}^m P_{mi}^2\lambda_i \end{pmatrix} \\ &= \frac{1}{n-1}\begin{pmatrix} \sum_{j=1}^m(\sum_{i=1}^m H_{i1}K_{ij})^2\lambda_j & \cdots & & \cdots \\ \vdots & \ddots & & \vdots \\ \cdots & \cdots & & \sum_{j=1}^m(\sum_{i=1}^m H_{im}K_{ij})^2\lambda_j \end{pmatrix}. \end{aligned} \tag{24}$$

$$(\boldsymbol{C}_{XH})_{ll} = \frac{1}{n-1}\sum_{j=1}^m (\boldsymbol{H}^T\boldsymbol{K})_{lj}^2\lambda_j = \frac{1}{n-1}\sum_{j=1}^m\left(\sum_{i=1}^m \boldsymbol{H}_{il}\boldsymbol{K}_{ij}\right)^2\lambda_j, \tag{25}$$

Since the values of $P_{ij}$ are not equal, the variance for each column of matrix $XH$ (which is the value on the diagonal of $C_{XH}$) is also not equal. Because the Hadamard transform is a fixed orthogonal transformation, a fixed orthogonal transformation cannot uniformly adjust the variance in all directions, resulting in the variance after transformation still being uneven.

## A.3 THE VARIANCE AFTER HADAMARD ROTATION IS STILL INCONSISTENT

We provide an example of a simple random 4x4 matrix after it has undergone the Hadamard transform. Example of a simple random 4x4 matrix R and a 4x4 Hadamard matrix H::

$$\boldsymbol{R} = \begin{pmatrix} 3 & -1 & 0 & -4 \\ -2 & 3 & -3 & 1 \\ 1 & -3 & 4 & -3 \\ -2 & 1 & -1 & 6 \end{pmatrix} \tag{26}$$

$$\boldsymbol{H} = \frac{1}{2} \begin{pmatrix} 1 & 1 & 1 & 1 \\ 1 & -1 & 1 & -1 \\ 1 & 1 & -1 & -1 \\ 1 & -1 & -1 & 1 \end{pmatrix} \tag{27}$$

Calculate the covariance matrix $C_{RH}$ of the matrix RH after the Hadamard transform:

$$\boldsymbol{C_{RH}} = \begin{pmatrix} \mathbf{1.83} & -4.83 & -2.83 & 2.00 \\ -4.83 & \mathbf{30.50} & 2.17 & -5.67 \\ -2.83 & 2.17 & \mathbf{8.75} & -1.50 \\ 2.00 & -5.67 & -1.50 & \mathbf{2.17} \end{pmatrix} \tag{28}$$

In the random example we provided, it can be seen that after the Hadamard rotation, there is a significant imbalance in the variance across the columns (which are the diagonal values of the covariance matrix $C_{RH}$).

We observe the distribution within the actual Mamba network. In the Vim-T network, we compare the variances before and after the Hadamard rotation for the output projection inputs of the 1st, 10th, and 20th blocks, as shown in Fig. 9. Additionally, Hadamard rotation is applied to the gate projection inputs of the 1st, 10th, and 20th blocks in Fig. 10, as well as to the gate projection weights of the 1st, 10th, and 20th blocks in Fig. 11.

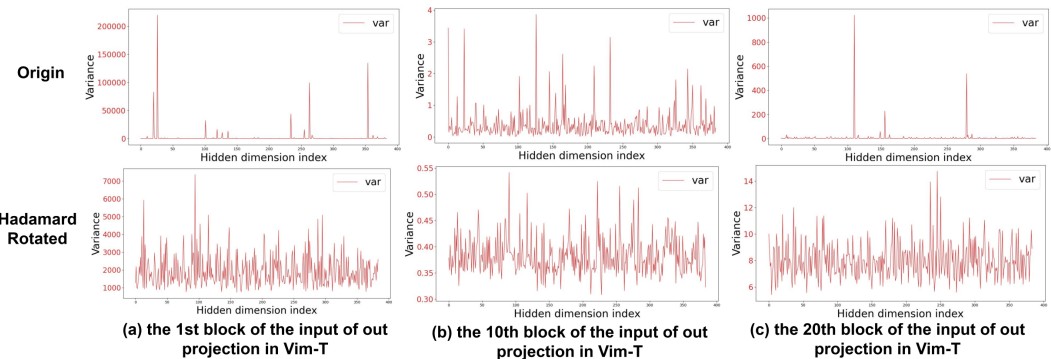

Figure 9: Variances of input of output projection layer blocks in Vim-T: unequal across channels pre- and post-Hadamard rotation

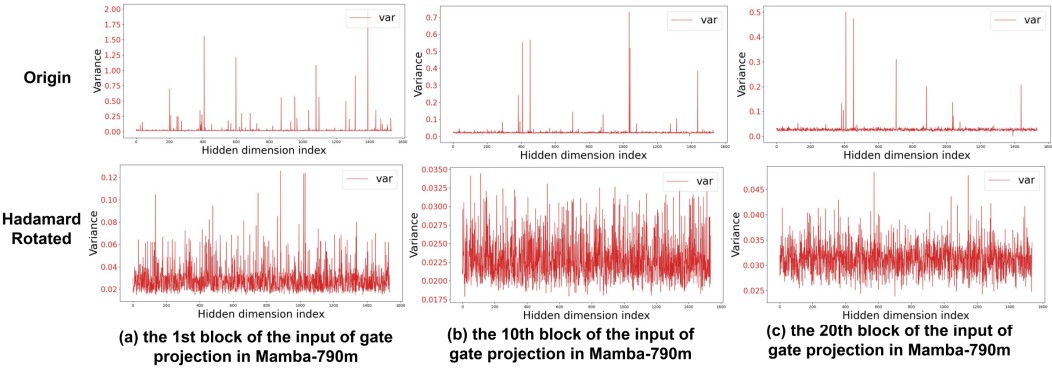

Figure 10: Variances of input of gate projection layer blocks in Mamba-790m: unequal across channels pre- and post-Hadamard rotation

### A.4 KLT-ENHANCED ROTATION TO ENSURE COLUMN VARIANCE BALANCE

**KLT rotation before Hadamard rotation.** We perform the KLT transformation on $X$ to obtain a new matrix $X'$, which is multiplying X by the matrix K on the right:

$$\boldsymbol{X'} = \boldsymbol{XK}, \tag{29}$$

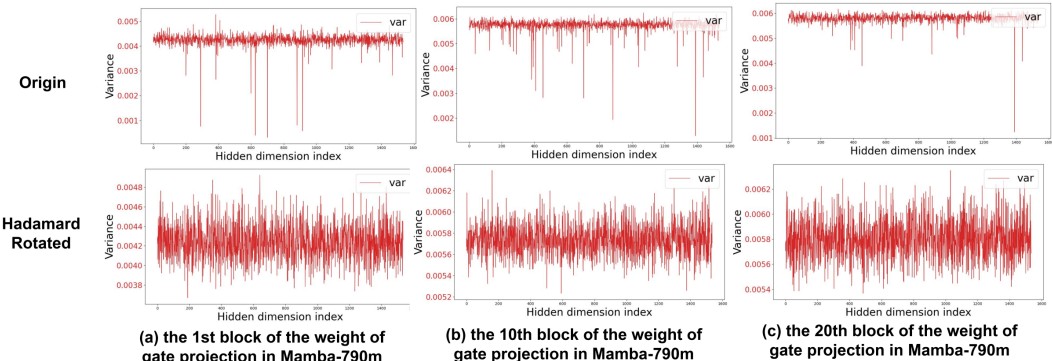

**Figure 11:** Variances of weight of gate projection layer blocks in Mamba-790m: unequal across channels pre- and post-Hadamard rotation

Equation 29 is the KLT transform. We calculate the covariance matrix of the data $X'$ as $C_{X'}$:

$$C_{X'} = \frac{1}{n-1} X'^T X',\tag{30}$$

We substitute Equation 29 into Eq 30:

$$C_{X'} = \frac{1}{n-1}(XK)^T(XK) = \frac{1}{n-1}K^T X^T X K,\tag{31}$$

In Eq 19, we have $X^T X = K\Lambda K^T$ which can be substituted into Equation 31.

$$C_{X'} = \frac{1}{n-1}K^T K\Lambda K^T K = \frac{1}{n-1}\Lambda,\tag{32}$$

We use the Hadamard matrix H to transform on $X'$ to obtain $X''$:

$$X'' = X'H,\tag{33}$$

Calculate the covariance matrix $C_{X''}$ of $X''$:

$$
\begin{aligned}
C_{X''} &= \frac{1}{n-1}(X'')^T X'' \\
&= \frac{1}{n-1}(X'H)^T X'H \\
&= \frac{1}{n-1}H^T X'^T X'H \\
&= \frac{1}{n-1}H^T C_{X'}H \\
&= \frac{1}{n-1}H^T \Lambda H.
\end{aligned}
\tag{34}
$$

Every element $h_{ij}$ of the Hadamard matrix is $1/\sqrt{m}$ or $-1/\sqrt{m}$: The composition of $\Lambda$ is $(\lambda_1, \lambda_2, ..., \lambda_n)$. Substitute Equation 21 into Equation 34.

$$C_{X''} = \frac{1}{n-1}\begin{pmatrix} \sum_{i=1}^m H_{i1}^2\lambda_i & \cdots & \cdots & \cdots \\ \cdots & \sum_{i=1}^m H_{i2}^2\lambda_i & \cdots & \cdots \\ \vdots & \vdots & \ddots & \vdots \\ \cdots & \cdots & \cdots & \sum_{i=1}^m H_{im}^2\lambda_i \end{pmatrix}\tag{35}$$

Substitute the actual value of $H_{ij}^2 = 1/m$, into Equation 35. Further calculate $C_{X''}$, and the result is:

$$C_{X''} = \frac{1}{(n-1)m}\begin{pmatrix} \sum_{i=1}^m \lambda_i & \cdots & \cdots & \cdots \\ \cdots & \sum_{i=1}^m \lambda_i & \cdots & \cdots \\ \vdots & \vdots & \ddots & \vdots \\ \cdots & \cdots & \cdots & \sum_{i=1}^m \lambda_i \end{pmatrix}\tag{36}$$

The main diagonal elements of Equation 36 represent the variance of each column of the matrix $X''$, which has been transformed first by KLT rotation and subsequently by Hadamard rotation. As evident from the equation, these variances are observed to be perfectly uniform across all columns.

## A.5 THE CALCULATION PROCESS OF INCREASING PARAMETERS AND ADDING COMPUTATIONAL LOAD

In this section, we take the Mamba-2.8b model as an example to illustrate the increase in parameter count and computational load. The Mamba-2.8b model comprises 64 blocks, and we assume a token quantity of 1024 for our calculations. We detail the computational overhead introduced by our smoothing process. The original parameter count of the Mamba-2.8b model is 2.8 billion. In our Method 4.3, we integrated a smoothing process by replacing the SiLU activation function with the S-SiLU activation function, which added 5,120 smoothing scale parameters. Additionally, we modified the mul operation to an addcmul operation, contributing an additional 16 parameters. Consequently, the percentage increase in parameter count is calculated as: .

$$\frac{((5120 + 16) \times 64)}{2.8 \times 10^9} \approx 0.01\%. \tag{37}$$

We estimate the original computational load of the Mamba-2.8b model to be 2.8 TFlops. In Method 4.2, we employed the online Hadamard technique, which has a complexity of $O(nlog_2(n))$. The complexity of the online Hadamard transformation is provided in Equation (1) of the paper by (Fino & Algazi, 1976). Within the Mamba block, we inserted an online Hadamard transformation of size [16,16] for the matmul smoothing and similarly used a [5120,5120] Hadamard transformation for the output projection smoothing. Thus, the percentage increase in computational load is calculated as:

$$\frac{(1024 \times 5120 \times 16 \times \log_2(16) + 1024 \times 5120 \times \log_2(5120)) \times 64}{2.8 \times 10^{12}} \approx 0.91\% \tag{38}$$

This analysis demonstrates that our proposed enhancements introduce minimal increases in both parameter count and computational load, making them practical for efficiency-critical applications.

## A.6 ADDITIONAL RESULTS

Table 5 shows the performance of increasing the KLT based rotate method and the smooth based rotate method on a variety of visual tasks.

Table 6 shows the performance of increasing the KLT based rotate method and the smooth based rotate method on a variety of language tasks.

Table 5: Add our methods sequentially on vision tasks

| Vision tasks | | Vim | | | | | Mamba-ND | | |
|---|---|---|---|---|---|---|---|---|---|
| Models | | **Vim-T** | Vim-T[†] | Vim-S | Vim-S[†] | Vim-B | Mamba-2d S | Mamba-2d B | Mamba-3d |
| | FP | 76.1 | 78.3 | 80.5 | 81.6 | 80.3 | 81.7 | 83.0 | 89.6 |
| | RTN | 37.4 | 32.4 | 68.8 | 72.8 | 52.2 | 79.9 | 82.2 | 88.9 |
| W8A8 | +KLT-Enhanced Rotation | 48.4 | 47.7 | 73.4 | 77.2 | 72.9 | 80.5 | 82.2 | 89.2 |
| | +Smooth-Fused Rotation | 75.6 | 77.8 | 80.3 | 81.4 | 80.1 | 81.5 | 82.8 | 89.2 |
| | RTN | 26.3 | 25.0 | 66.1 | 70.0 | 46.2 | 40.6 | 78.8 | 87.1 |
| W4A8 | +KLT-Enhanced Rotation | 41.7 | 38.9 | 71.3 | 75.7 | 71.2 | 79.2 | 81.7 | 88.0 |
| | +Smooth-Fused Rotation | 72.1 | 73.7 | 79.4 | 80.4 | 79.6 | 80.4 | 82.5 | 88.4 |

## A.7 GENERALIZATION EVALUTATION OF THE KLT-ENHANCED ROATION

To generate $K$ without introducing online computational overhead, we use the calibration method, which is a common practice of PTQ works (Lin et al., 2023; Xiao et al., 2022; Shao et al., 2023). In most cases, this method can effectively characterize the whole data distribution by a small subset to achieve rapid quantization optimization.

For instance, the experiments shown in Table 2 are calibrated from the HellaSwag dataset. Zero-shot evaluation results, especially on other datasets like ARC-E, ARC-C, PIQA, and Winogrande can effectively demonstrate the generalization ability.

Furthermore, for various inputs, we visualize the activations distribution of the in-projection layer of the first block in the Vim-T model. Then we respectively performs the classic Hadamard rotation and our KLT-enhanced rotation.

Table 6: Add our methods sequentially on language tasks

| Models | Bit Width | Mehods | Avg ACC | Arc-E | Arc-C | PIQA | WinoGrande | HellaSwag |
|---|---|---|---|---|---|---|---|---|
| Mamba-130m | FP16 | - | 44.7 | 48.0 | 24.3 | 64.5 | 51.9 | 35.3 |
| | W8A8 | RTN | 41.5 | 41.0 | 25.4 | 56.3 | 51.7 | 32.9 |
| | | +KLT-Enhanced Rotation | 42.2 | 42.3 | 25.1 | 59.4 | 50.4 | 33.9 |
| | | +Smooth-Fused Rotation | 43.9 | 45.9 | 24.2 | 62.5 | 52.3 | 34.7 |
| | W4A8 | RTN | 35.2 | 27.6 | 22.4 | 49.8 | 51.0 | 25.1 |
| | | +KLT-Enhanced Rotation | 39.0 | 33.5 | 23.9 | 55.8 | 51.7 | 30.1 |
| | | +Smooth-Fused Rotation | 39.8 | 36.2 | 24.6 | 56.9 | 50.9 | 30.5 |
| Mamba-370m | FP16 | - | 50.9 | 55.1 | 28.0 | 69.5 | 55.3 | 46.5 |
| | W8A8 | RTN | 45.7 | 43.8 | 27.5 | 59.6 | 53.5 | 44.0 |
| | | +KLT-Enhanced Rotation | 49.0 | 50.8 | 28.9 | 66.1 | 53.7 | 45.6 |
| | | +Smooth-Fused Rotation | 50.0 | 54.1 | 28.2 | 67.3 | 53.8 | 46.5 |
| | W4A8 | RTN | 36.3 | 27.2 | 25.7 | 50.6 | 50.7 | 27.4 |
| | | +KLT-Enhanced Rotation | 43.1 | 37.0 | 27.0 | 60.1 | 51.6 | 39.7 |
| | | +Smooth-Fused Rotation | 44.1 | 38.7 | 27.3 | 61.5 | 53.9 | 39.0 |
| Mamba-790m | Fp16 | - | 54.8 | 61.2 | 29.5 | 72.1 | 56.0 | 55.1 |
| | W8A8 | RTN | 44.2 | 43.7 | 26.1 | 60.7 | 52.7 | 37.8 |
| | | +KLT-Enhanced Rotation | 51.3 | 52.4 | 32.3 | 65.1 | 55.0 | 51.9 |
| | | +Smooth-Fused Rotation | 53.8 | 59.5 | 30.6 | 68.9 | 55.8 | 54.6 |
| | W4A8 | RTN | 35.4 | 28.0 | 25.0 | 50.7 | 49.6 | 23.5 |
| | | +KLT-Enhanced Rotation | 42.3 | 33.6 | 27.1 | 58.0 | 53.5 | 39.2 |
| | | +Smooth-Fused Rotation | 45.8 | 39.8 | 30.2 | 62.5 | 52.3 | 44.3 |
| Mamba-1.4b | FP16 | - | 58.6 | 65.5 | 32.8 | 74.2 | 61.4 | 59.1 |
| | W8A8 | RTN | 53.9 | 54.9 | 31.4 | 69.4 | 56.8 | 57.0 |
| | | +KLT-Enhanced Rotation | 56.1 | 60.8 | 32.7 | 71.3 | 57.9 | 57.9 |
| | | +Smooth-Fused Rotation | 58.3 | 64.9 | 32.8 | 73.8 | 60.9 | 59.1 |
| | W4A8 | RTN | 51.6 | 51.7 | 30.9 | 64.9 | 57.4 | 53.0 |
| | | +KLT-Enhanced Rotation | 54.3 | 57.8 | 30.6 | 70.4 | 57.0 | 55.9 |
| | | +Smooth-Fused Rotation | 54.3 | 57.7 | 31.3 | 70.3 | 56.7 | 55.3 |
| Mamba-2.8b | FP16 | - | 62.2 | 69.7 | 36.3 | 75.2 | 63.5 | 66.1 |
| | W8A8 | RTN | 57.5 | 60.4 | 33.8 | 71.0 | 58.5 | 63.9 |
| | | +KLT-Enhanced Rotation | 60.5 | 68.3 | 35.3 | 73.9 | 60.2 | 64.7 |
| | | +Smooth-Fused Rotation | 62.1 | 69.8 | 36.3 | 75.5 | 62.9 | 66.0 |
| | W4A8 | RTN | 53.7 | 54.5 | 32.6 | 66.2 | 56.7 | 58.6 |
| | | +KLT-Enhanced Rotation | 58.0 | 60.7 | 35.8 | 71.6 | 60.7 | 61.1 |
| | | +Smooth-Fused Rotation | 58.5 | 62.2 | 35.5 | 72.0 | 60.9 | 62.1 |

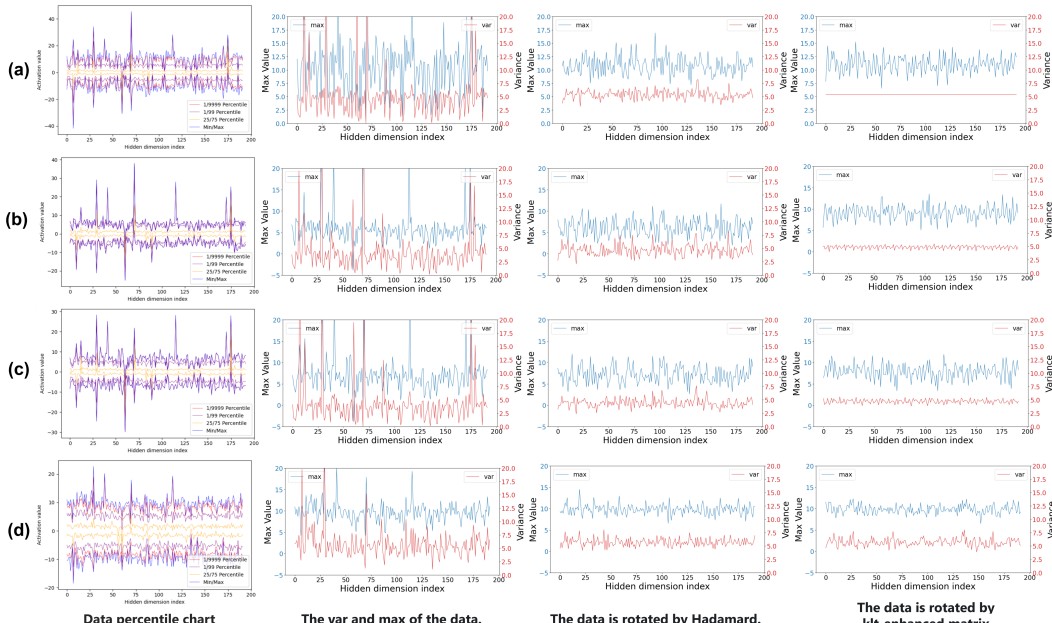

Figure 12: Analysis of Activation Values in the in_projection of the block 0 of the Vim-T model.(a) Display using 384 images from ImageNet for calibration. (b), (c) and (d) Display of other data outside the calibration set.

Figure 12(a) shows the calibration data and its distribution after KLT-enhanced rotation, while Figure 12(b), (c), and (d) display the situation for non-calibration data. Figure 12(b) and (c) clearly demonstrate that the KLT-enhanced rotation can also effectively maintain the uniformity of the

maximum values and variances of data with similar distribution to calibration. Its effect is significantly better than using only Hadamard rotation. In most cases, the calibration of PTQ can sufficiently characterize various data distributions. For extreme circumstances where the distribution of the inputs are dissimilar to that of the calibration (as shown in Figure 12(d)), the effect of the KLT-enhanced matrix is still not worse than that of Hadamard rotation.

## A.8   COMPARISON OF NUMERICAL DISTRIBUTIONS BETWEEN VIT AND VIM MODELS

The distributions of Transformers and Mamba are significantly different. For instance, we randomly sample 96 images from ImageNet, then feed them into the classic Vit (Alexey, 2020) model and Vim (Xu et al., 2024) model. Next, we calculate the top-3 channel maximums and top-3 channel variances of the input activations of all quantized modules in the last block. It can be clearly seen that Vim has more uneven distribution and more outliers than Vit, leading to great challenges for quantization.

Table 7: Statistics of the top 3 maximum values and maximum variances among activation channels. We choose the modules to be quantized of the last block of Vim and Vit. Data are randomly sampled from the ImageNet dataset.

| Model | Module | Top3 Channel Maximums | Top3 Channel Variances |
|---|---|---|---|
| vit-base-patch16-224 | attention.qkv_proj | 4.7 / 3.8 / 3.7 | 0.4 / 0.4 / 0.4 |
| | attention.qk_matmul.q | 7.1 / 7.1 / 7.1 | 2.1 / 1.6 / 1.6 |
| | attention.qk_matmul.k | 10.8 / 10.6 / 10.5 | 4.5 / 4.4 / 4.3 |
| | attention.o_proj | 11.6 / 8.8 / 8.6 | 4.1 / 3.4 / 3.3 |
| | attention.pv_matmul.p | 4.8 / 4.7 / 4.2 | 0.1 / 0.1 / 0.1 |
| | attention.pv_matmul.v | 14.9 / 12.5 / 12.5 | 6.5 / 5.4 / 4.5 |
| | mlp.fc1 | 11.4 / 8.2 / 7.8 | 2.3 / 2.0 / 1.6 |
| | mlp.fc2 | 17.26 / 13.15 / 13.01 | 12.10 / 9.84 / 7.35 |
| vim-base-patch16-224 | in_proj | **49.3 /48.8 / 38.9** | **230.8 / 95.5 / 62.8** |
| | conv1d | **48.2 / 36.2 / 36.2** | **41.3 / 27.8 / 27.7** |
| | x_proj | 15.1 / 12.0 / 11.1 | 8.6 / 4.5 / 3.9 |
| | dt_proj | 14.9 / 13.6 / 4.1 | 11.0 / 10.9 / 10.6 |
| | matmul_in1 | **75.2 / 59.0 / 58.2** | **20.8 / 19.84 / 18.4** |
| | matmul_in2 | 8.5 / 6.7 / 5.4 | 1.8 /1.6 / 1.6 |
| | out_proj | **1371.6 / 1064.3 / 930.3** | **8854.9 / 2513.8 / 1377.8** |

## A.9   VISUALIZED AMPLIFICATION EFFECT OF THE PSCAN OPERATOR

In Section 1, it is stated that the Parallel Scan (PScan) further amplifies the outliers of activations. Despite the corresponding explanation, we here provide the visualized data to further support this viewpoint. Specifically, we focus on the activation statistics of the Vim-T network on the ImageNet dataset and visually present the distribution differences between the inputs and outputs of Pscan in different blocks.

Figure 13(a) presents the distribution of activation values input of Pscan, which corresponds to $\overline{\boldsymbol{B}}\boldsymbol{x}(t)$ in Equation 5. Meanwhile, Figure 13(b) illustrates the distribution of activation values subsequent to the application of the Pscan operator. It has been observed that following the application of the pscan operator, the distribution differences of activation values in the hidden dimension become more pronounced. This phenomenon can be attributed to the multiple consecutive multiplications involved in the internal implementation of the pscan operator.

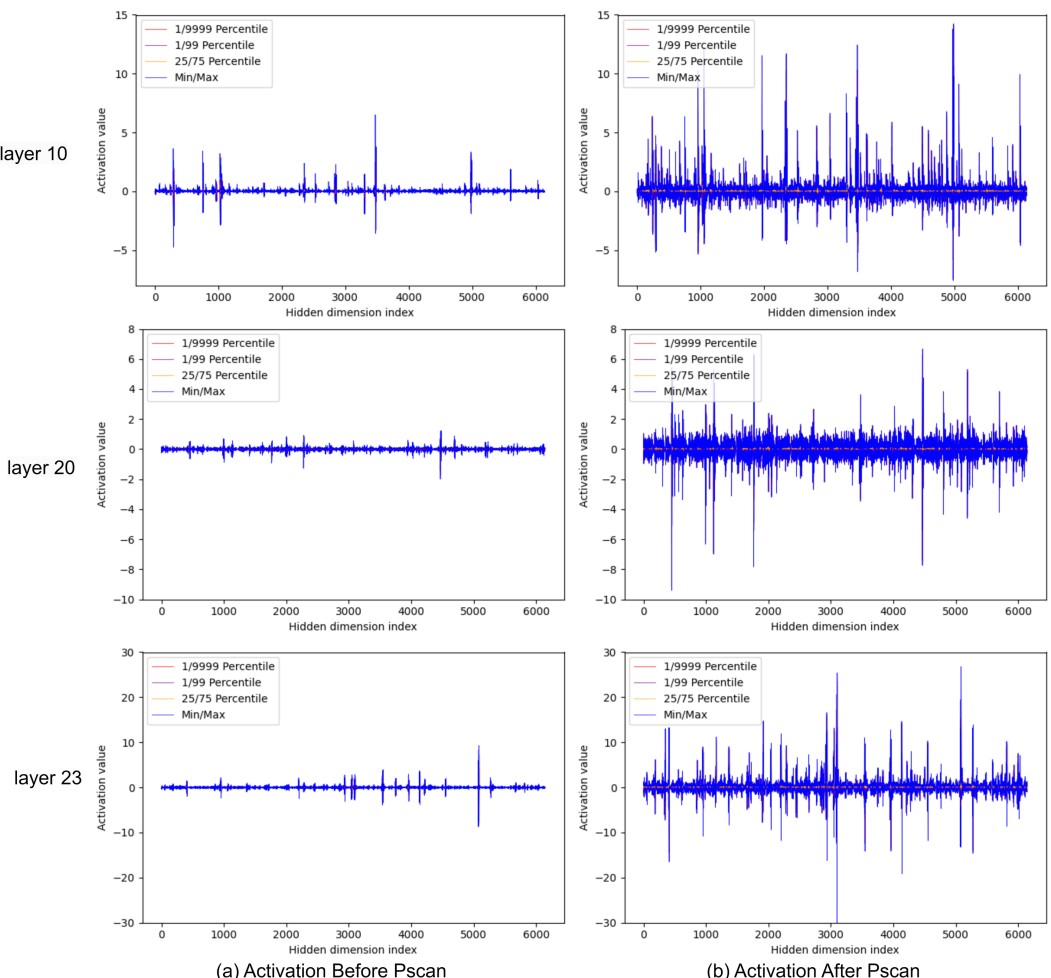

Figure 13: The amplification phenomenon of activations by the Pscan operator in different blocks of the Vim-T model.

