# OpenReview forum: "MambaQuant: Quantizing the Mamba Family with Variance Aligned Rotation Methods"
_ICLR.cc/2025/Conference — ICLR 2025 Poster_

### Official Review · Reviewer_kvr1 · 2024-10-31

**Soundness:** 3
**Presentation:** 3
**Contribution:** 3
**Rating:** 8
**Confidence:** 3

**Summary:**

The authors propose a post-training quantization framework, called MambaQuant, which applies to Mamba models. MambaQuant relies on the observation that there are weights and outliers in Mamba models which are significantly larger than the rest, and that this disparity is increased by the parallel scan operation. The authors first study the effectiveness of the Hadamard transformation, which incorporates pre- and post- multiplies the features by Hadamard matrices so as to spread out the largest values, and show that it is ineffective at equalizing variance across channels, which makes quantization performance suffer. Then they discuss the MambaQuant framework which consists of two operations:
- An offline whitening step, which they call the Karhunen-Loeve transform: first diagonalize the feature covariance matrix at the given layer, then multiply the appropriate Hadamard matrix by the matrix of eigenvectors of the feature covariance. Multiplying the features by this product matrix instead of the Hadamard matrix will spread out the massive values and also ensure that the variance across each channel is balanced.
- An online smoothing step: do a transformation on the SILU, e.g., $x \sigma(W x) \mapsto x \sigma(s \odot [(W \oslash s)x])$ where $\odot, \oslash$ are elementwise multiplication/division respectively and $s$ is a smoothing vector, and there exists an algorithm to compute $(W \oslash s)x$ accurately and efficiently via fusing with multiple operations.

**Strengths:**

- The proposed toolkit MambaQuant is clearly motivated, and the motivation follows from both theoretical and empirical perspectives; the weaknesses of the prior methodology (that is, smoothing large values via Hadamard transformation) are clearly demonstrated in the context of Mamba.
- Accordingly, the solutions proposed by MambaQuant are shown to resolve these issues effectively using the same methodology.
- The shown empirical results demonstrate minimal performance degradation compared to Mamba on both language and vision tasks.
- The paper conducts several ablations to show that in their experimental setup, all changes significantly improve a property of the model to make quantization easier.

**Weaknesses:**

- The whitening transformation (KLT) seems to be computationally prohibitive to run. It is true that the paper thus classifies it as an offline transformation. However, it is still not clear whether the eigenvectors of the feature matrix are similar across batches (so that the transform truly whitens the data and equalizes variance across all channels). That would be an interesting insight, if true, but it's not confirmed here, and so the KLT needs more empirical evidence on more diverse data to be validated in real applications.
- The methods proposed are only studied in the context of, and only apply to, the Mamba family of architectures. This is an explicit limitation of the paper. It would be great if the methods here transferred to different architectures without significant performance downgrades (i.e., they still sort of work).

**Questions:**

- See weaknesses section; more empirical validation would be great in order to demonstrate the generalizability of the MambaQuant operations.
- Is there any aspect of runtime which is significantly improved or made worse by these changes? E.g. prefill time or average runtime? Same with the memory constraints. It's acknowledged that the appendix contains some estimates for time and space complexity but it would be interesting to see how these bear out in practice.

---

> ### Author Response · Authors · 2024-11-22
> **Response to Reviewer kvr1**
>
> We sincerely thank you for your valuable time and efforts in reviewing our manuscript, and we tried our best to address all your concerns as follows.
>
> > The KLT needs more empirical evidence on more diverse data.
>
> Thanks for this constructive comment. To evaluate the robustness of our KLT method, we perform the following experiments.
>
> 1. In Table 2 of the manuscript, we evaluate our MambaQuant on Mamba-LLM. The calibration data is sampled from the HellaSwag dataset. Zero-shot evaluation results, **especially on other datasets** like ARC-E, ARC-C, PIQA, and Winogrande, can effectively demonstrate the generalization ability of our KLT method.
>
> 2. We have added **visualized examples in Figure 12 of Appendix A.7**. It can be clearly seen that: For inputs outside the calibration set and with similar distribution, our KLT-enhanced rotation can also achieve significant optimization; For inputs outside the calibration set but with distinct distribution, the effectiveness of the KLT-enhanced rotation is still not worse than the Hadamard rotation.
>
> 3. It should be mentioned that it is a common practice for PTQ to characterize a wide range of data distribution with a static subset, which works effectively in most cases.
>
> > The methods proposed are only studied in the context of, and only apply to, the Mamba family of architectures. It would be great if the methods here transferred to different architectures.
>
> First of all, this work focuses on quantization for the **Mamba family models**.
>
> Second, our methods also can be **well generalized to other families** such as the mainstream Transformer-based Large Language Models (T-LLMs). The inconsistent variance problem of the Hadamard rotation arises from the algorithm itself and does not depend on any specific model structure. Despite the imperceptible effect on T-LLMs, this limitation is particularly significant when quantizing the Mamba models.
>
> 1. For the KLT-enhanced rotation, it essentially changes the rotation matrix from $H$ to $KH$ as stated in the Equation (11). Considering that the rotation scheme is quite mature for T-LLMs [1,2], our KLT-enhanced rotation can be directly applied.
>
> 2. For the Smooth-fused rotation, it uses the smoothing technique to uniform the channel variances before the online rotation. We further integrate the extra smoothing factors into weights with consideration of the Mamba structure. Since both the smoothing methods [3,4] and rotation methods are well-developed for T-LLMs, our smooth-fused rotation can also be easily applied.
>
> Lastly, we further explore the robustness of the algorithms in MambaQuant on T-LLMs. As shown in the table below, we quantize llama2-7b to w4a4 using the algorithms of Quarot and MambaQuant respectively. It can be clearly concluded that the **KLT-enhanced rotation also outperforms the SOTA baseline**.
>
> | Method | Setting | ARC-c | HellaS | PIQA  | Avg   |
> | :------: | :-------: | :-----: | :------: | :-----: | :-----: |
> | -      | FP16    | 46.42 | 75.9   | 79.16 | 66.16 |
> | Quarot | W4A4    | 42.32 | 72.91  | 77.20 | 64.14 |
> | Ours   | W4A4    | **44.13** | **74.34**  | **78.21** | **65.56** |
>
> > Is there any aspect of runtime which is significantly improved or made worse by these changes? It would be interesting to see how these bear out in practice.
>
> Thanks for this constructive suggestion.We compared the average inference speeds before and after quantization based on the Mamba-2.8b model with seqlen 512 in A6000. Under w8a8 quantization setting, the acceleration ratio is 1.2 times. Due to the limited rebuttle time, our kernel has not been highly optimized, and there is still room for improvement in speed.
>
> |            | speed  |        |         |memory      |      |               |
> | :----------: | :------: | :------: | :-------: | :----: | :----: | :-------------: |
> |            | fp16   | w8a8   | Speedup | fp16 | w8   | Memory saving |
> | mamba-2.8b | 65.9ms | 54.1ms | 1.22x   | 5.3G | 2.8G | 1.89x         |
>
> If there are still any unresolved doubts, please feel free to let us know, and we will make every effort to solve them.
>
> **References:**
>
> [1] Quarot: Outlier-free 4-bit inference in rotated llms.
>
> [2] SpinQuant--LLM quantization with learned rotations.
>
> [3] Smoothquant: Accurate and efficient post-training quantization for large language models.
>
> [4] Omniquant: Omnidirectionally calibrated quantization for large language models.

---

> > ### Comment · Reviewer_kvr1 · 2024-11-23
> > **Response to Rebuttal**
> >
> > Thanks for the detailed response. Since all weaknesses and questions were addressed (Appendix A.7 and the results showing generalizability to transformers are especially convincing), I will raise my score.

---

> ### Author Response · Authors · 2024-11-25
>
> Dear Reviewer,
>
> Thank you for your constructive comments and valuable suggestions.
>
> We are glad to address all your questions and sincerely appreciate your recognition.
>
> Best, Authors

---

### Official Review · Reviewer_AGfn · 2024-11-01

**Soundness:** 2
**Presentation:** 3
**Contribution:** 3
**Rating:** 6
**Confidence:** 3

**Summary:**

The paper explores post-training quantization for Mamba family of state-space models and identifies several key challenges, namely the presence of outliers in gate projections, output projections and the fact that Mamba’s PScan operation further amplifies these outliers.

Authors proposed MambaQuant, a post-training quantization framework for Mamba model family, consisting of two extensions over Hadamard-based methods: 1) Karhunen-Loève Transformation (KLT) enhanced rotation; and 2) Smooth-fused rotation, which equalizes channel variances and can be merged into model weights.

Experiments show that MambaQuant can quantize both weights and activations into 8-bit with less than 1% accuracy loss for Mamba-based vision and language tasks.

**Strengths:**

* Authors demonstrate the effectiveness of MambaQuant on both vision and language tasks and different model sizes. Solid performance against the recent PTQ methods including GPTQ, SmoothQuant, QuaRot.
* Ablation studies and figures to demonstrate that how the proposed method improves the distribution of problematic activations and hence improves their quantizability.
* Authors intend to release the code.

**Weaknesses:**

* Paper claims to be the first comprehensive analysis of quantizability of Mamba, however many of the insights including the existence of outliers in gate and output projections were already discussed in “Mamba-PTQ” [1]. Could you elaborate on any additional differences between these two approaches?

* It seems that KLT enhanced rotation is very straightforward and boils down to simply applying an SVD decomposition on weights/activations (or equivalently, eigenvalue decomposition on the covariance matrix), and fusing the emerging matrix K into H. Is there is any additional insight to it?

[1] Pierro et al., "Mamba-PTQ: Outlier Channels in Recurrent Large Language Models". ArXiV:2407.12397

**Questions:**

* Would be great if authors could include detailed per-task results for LM-eval benchmark in the appendix, to improve reproducibility and make it easier to compare to.

---

> ### Author Response · Authors · 2024-11-22
> **Response to Reviewer AGfn**
>
> We sincerely thank you for your valuable time and efforts in reviewing our manuscript, and we tried our best to address all your concerns as follows.
>
> > Could you elaborate on any additional differences between Mamba-PTQ and your work?
>
> When we were writing our paper, we surveyed mainstream AI conferences and journals but did not find Mamba-PTQ [1]. Up to now, we note that this preprint has not yet passed the review of professional peers.
>
> Based on your suggestion, we have carefully reviewed Mamba-PTQ. The most closely related aspect is its discussion of outliers in Mamba networks. While both works identify the existence of outliers, our study goes a step further by revealing that the unique PScan operator in Mamba networks can amplify these outliers. More importantly, **our paper offers several unique contributions, which we outline below**:
>
> 1. **Variance viewpoint**. We analyze not only the issue of outliers, but also the importance of the consistent channel variances for quantization. Balanced variance is essential for reducing quantization error but typically overlooked by existing works.
>
> 2. **KLT-enhanced rotation**. We theoretically analyze why existing advanced rotation[4,5] techniques fail to address the issue of variance inconsistency (see details in Section 4.1) and propose a variance-consistent KLT-enhanced rotation technique (see details in Section 4.2).
>
> 3. **Smooth**-**Fused scheme**. As metioned In Section 4.3, our Smooth-fused scheme  integrates Smooth parameters into model weights during the offline phase, avoiding any additional computation during online inference.   In contrast, Mamba-PTQ adopts traditional online smoothing which introduces additional computational overhead.
>
> 4. **MatMul quantization**. Mamba-PTQ only quantizes linear layers, while our work further quantizes the MatMul operator. This operation is hard to be quantized as effected by the PScan and also consumes large inference latency especially for the prefill phase.
>
> 5. **Superior performance**. Our work achieves nearly lossless quantization accuracy (less than 1%) in W8A8 setting on both computer vision and large language tasks. In contrast, Mamba-PTQ does not report its performance on computer vision tasks, and its quantization incurs significant accuracy loss on large language tasks. For instance, the quantization accuracy of Mamba-PTQ in W8A8 drops by more than 10%, with the Mamba-790 model experiencing a drop of over 20%.
>
> > It seems that KLT enhanced rotation is very straightforward. Is there is any additional insight to it?
>
> Obtaining the matrix K is not theoretically complex as described in your comment (eigenvalue decomposition on the covariance matrix). Although, we would like to stress the motivation and the effectiveness of this approach besides its computational process.
>
> 1. Through detailed analysis in Section 4.1, we reveal the shortcoming of the current rotation methods. They do not take the matter of consistent variances into consideration, leading to sub-optimal performance for the Mamba family. We thus introduce the method of KLT, which has not been adopted before to solve this problem.
>
> 2. To provide readers with a clearer insight into the effectiveness of our KLT method, we present Equation (8) with (12), and Equation (9) with (13) in a comparative way. As shown in Equation (13) of the manuscript, this method is simple but can effectively equalize the channel variances.
>
> 3. The rotation scheme [4,5] has not been applied to the Mamba structures before. We effectively apply it to the Mamba structure.
>
> In conclusion, the KLT method is **practically simple and also significantly effective** for quantization, promoting the deployment of the Mamba family models. **Our contribution is explaining why it works and how it works**.
>
> > Would be great if authors could include detailed per-task results for LM-eval benchmark in the appendix, to improve reproducibility and make it easier to compare to.
>
> We have included all per-task results for LM-eval benchmark in Table 6 in the Appendix (A.6). We also plan to open source all related codes and data to promote the development of the community.
>
> If there are still any unresolved doubts, please feel free to let us know, and we will make every effort to solve them.
>
> **References:**
>
> [1] Mamba-PTQ: Outlier Channels in Recurrent Large Language Models.
>
> [2] Smoothquant: Accurate and efficient post-training quantization for large language models.
>
> [3] Omniquant: Omnidirectionally calibrated quantization for large language models.
>
> [4] Quarot: Outlier-free 4-bit inference in rotated llms.
>
> [5] SpinQuant--LLM quantization with learned rotations.

---

> > ### Author Response · Authors · 2024-12-02
> >
> > Dear Reviewer AGfn,
> >
> > We hope this message finds you well. We would like to sincerely thank you for your thoughtful review and valuable feedback on our paper. Your positive assessment and constructive comments have been instrumental in helping us improve our work.
> >
> > If you find that our Responses have satisfactorily addressed your concerns, we would be grateful if you could consider reflecting this in your final assessment.
> >
> > Thank you again for your valuable contributions to our research.
> >
> > Warm regards,
> >
> > The Authors

---

### Official Review · Reviewer_5nSC · 2024-11-04

**Soundness:** 2
**Presentation:** 3
**Contribution:** 3
**Rating:** 5
**Confidence:** 4

**Summary:**

This paper find out current PTQ methods on Transformer-based LLMs failed to quantize Mamba. Then it proposes MambaQuant with offline KLT-enhanced rotation and online smooth-fused rotation to improve PTQ performance. Experiments show MambaQuant achieves 1\% accuracy loss on W8A8, better than QuaRot and SmoothQuant.

**Strengths:**

1. This paper studies the PTQ problems for Mamba,  finding out that current PTQ methods on Transformer-based LLMs failed to quantize Mamba.
2. MambaQuant applies SmoothQuant's smoothing vector to Mamba's non-linear SiLU and PScan.
3. Experiments show MambaQuant achieves 1\% accuracy loss on W8A8.

**Weaknesses:**

1. This paper does not detail how to get $K$ for different input $X$.
2.  KLT-enhanced ratation seems unable to solve all input's channel in-variance and outliers.

**Questions:**

KLT-enhanced rotation $KH$ seems to be also fixed, like Hadmard matrix $H$, for different input. To perform offline transformation, eigenvectors $K$ also need to be fixed for all input.  Thus it may not uniformly transform each X's channel variance?

---

> ### Author Response · Authors · 2024-11-22
> **Response to Reviewer 5nSC Part1**
>
> We sincerely thank you for your valuable time and efforts in reviewing our manuscript, and we tried our best to address all your concerns as follows.
>
> > This paper does not detail how to get K for different input X.
>
> As stated in line 263 in Section 4.2 of the manuscript, K is generated from calibration data. We have also added explicit statements in the Section 4.2 of the manuscript to stress this point. This statically generated $K$ can be well-generalized for different input $X$ .

---

> > ### Author Response · Authors · 2024-11-22
> > **Response to Reviewer 5nSC Part2**
> >
> > > KLT-enhanced rotation seems unable to solve all input's channel in-variance and outliers. KLT-enhanced rotation KH seems to be also fixed, like Hadamard matrix H, for different input. To perform offline transformation, eigenvectors K also need to be fixed for all input. Thus it may not uniformly transform each X's channel variance?
> >
> > We here respond to your concerns in Weakness 2 and Questions. The KLT-enhanced rotation can uniformly transform the channel variances for most inputs as explained below.
> >
> > (1) General explanation.
> >
> > The KLT rotation matrix$ $KH$ is **relevant to data distribution** rather than fixed to $\pm 1$ like the Hadamard rotation matrix H, thus giving chances to align channel variances. This process corresponds to the Equation (11) of the manuscript:
> >
> > $$P=KH.$$
> >
> > In detail, the K in Equation (11) is determined by the KLT's input data, which is a subset of all input. We would like to clarify that our KLT-enhanced rotation is **of great generalization ability to handle this gap** with the following analysis, experimental evaluation, and visualized data.
> >
> > (2) Further analysis.
> >
> > The KLT-enhanced rotation can also fit all data by inputting the online data to KLT. In this case, it can strictly equalize all channel variances as proved in the manuscript. Considering the extra online operation of eigenvalue decomposition and matrix multiplication that can increase the inference latency, we use the calibration method to perform the KLT in the offline mode.
> >
> > 1. Considering that the calibration data of PTQ can sufficiently characterize various distributions in most cases, the distributions between the calibration data and all input data can be regraded similar. Therefore, the KLT-enhanced rotation is also able to **effectively uniform the channel variances, just not strictly the same**.
> >
> > 2. For extreme circumstances where the distribution of the inputs are dissimilar to that of the calibration, the KLT-enhanced rotation may not be able to smooth the variances, but it still remains the ability to uniform the maximum values as the property of the Hadamard rotation. This is because the $KH$ rotation to $X$ can be seen as the $H$ rotation to $XK$.
> >
> > (3) The nature of PTQ.
> >
> > Our method belongs to the field of PTQ, whose essence is to achieve rapid optimized quantization through **characterizing the whole data distribution by** **a** **subset**. It is a common practice for PTQ and widely adopted by most of the PTQ works, such as AWQ [1], SmoothQuant [2], OmniQuant [3].
> >
> > Another kind of approach, Quantization Aware Training (QAT), leverages the complete training data and executes the whole training process. These methods are not that practical due to data privacy issues and expensive training costs.
> >
> > There are also studies on advancing the quality of the PTQ calibration set, such as selecting an appropriate calibration set and optimizing the generalization of the calibration set. However, it is not the focus of this work.
> >
> > (4) **Experimental evaluation of generalization ability**.
> >
> > In Table 2 of the manuscript, we evaluate our MambaQuant on Mamba-LLM. The calibration data is sampled from the HellaSwag dataset. Zero-shot evaluation results, especially on other datasets like ARC-E, ARC-C, PIQA, and Winogrande(detailed results for each task can be viewed in Table 6 of Appendix A.6), can effectively demonstrate the generalization ability of our KLT method.
> >
> > (5) **Visualized** **generalization ability**.
> >
> > We have added visualization examples in Figure 12 of Appendix A.7. It can be clearly seen that:
> >
> > 1. In most cases, inputs outside the calibration set and with similar distribution, our KLT-enhanced rotation can also achieve significant optimization.
> >
> > 2. In a few cases, inputs outside the calibration set but with distinct distribution, the effectiveness of the KLT-enhanced rotation is still not worse than the Hadamard rotation.
> >
> > (6) How can KLT-enhanced rotation address outliers.
> >
> > Through multiplying X with K, the distribution of X is changed but not variances-aligned yet. By multiplying H, the XK is endowed with uniformed maximum values (**owing to the property of Hadamard matrix**). At the same time, due to the transformed distribution of XK,  its channel variances are also aligned by multiplying H (as detailedly proved in the manuscript).
> >
> > If there are still any unresolved doubts, please feel free to let us know, and we will make every effort to solve them.
> >
> > **References:**
> >
> > [1] AWQ: Activation-aware Weight Quantization for On-Device LLM Compression and Acceleration.
> >
> > [2] Smoothquant: Accurate and efficient post-training quantization for large language models.
> >
> > [3] Omniquant: Omnidirectionally calibrated quantization for large language models.

---

> > ### Comment · Reviewer_5nSC · 2024-11-26
> >
> > Thanks for authors' reply. I still can not get how to generate K from calibration data. Can you give an example for a calibration set with n samples [$X_1, X_2, X_3, ......, X_n$] to obtain one single K.

---

> ### Author Response · Authors · 2024-11-26
>
> Sure, here is a detailed example showing how to obtain a single $K$ from the calibration dataset.
>
> **Step 1:**
>
> Supposing that the shape of each activation sample $X_i$ is (L, D)  (where L is the sequence length and D is the hidden dimension), the calibration dataset [$X_1$, $X_2$, $X_3$, ..., $X_N$] can be seen as a tensor with shape (N * L, D). This tensor corresponds to the character $X$ in the Equation (10) of the manuscript.
>
> **Step 2:**
>
> The Equation (10) first calculates the covariance matrix ${C}_X$ of $X\in \mathbb{R}^{NL,D}$, which can be written as:
> $$ {C}_X=\frac{1}{NL-1}{X}^T{X},$$
> where the covariance matrix $C_X$ has shape (D, D).
>
> **Step 3:**
>
> The Equation (10) then performs eigenvalue decomposition (ED) to $C_X$:
> $$\text{ED}(C_X)=\frac{1}{NL-1}{K\Lambda} {K}^T.$$
> Thereby, a single $K$ with shape (D, D) that can be obtained from the calibration dataset with N samples. This $K$ is actually the eigenvector matrix of the covariance matrix $C_X$ of the calibration data $X$.
>
> **More explanation:**
>
> The essence of the KLT method is to perform ED to the covariance matrix. The tensor shape of the covariance matrix is irrelevant to the number of samples (i.e., it is always (D, D)), owing to the calculation of $X^T X$. Thereby, we can always generate a single $K$ from the calibration dataset.

---

> > ### Author Response · Authors · 2024-12-02
> >
> > Dear Reviewer 5nSC,
> >
> > We hope this message finds you well.
> >
> > As the deadline for review updates is approaching, we are eager to know whether our answers well address your concerns, as it is crucial for us to have a candid and thorough discussion to continuously strengthen our method. Please share your thoughts on viewing our reply. We hope to resolve your doubts with our best efforts.
> >
> > Thank you again for your time and for sharing your valuable insights. We deeply appreciate your support.
> >
> > Warm regards,
> >
> > The Authors

---

### Official Review · Reviewer_KhMz · 2024-11-08

**Soundness:** 3
**Presentation:** 3
**Contribution:** 3
**Rating:** 6
**Confidence:** 3

**Summary:**

This paper proposes a quantization scheme for a family of under-explored models, mamba based models. The authors observe that the outliers exists in certain type of projections in mamba models, and the parallel scan further make the issue worse. And the Hadamard transform, commonly used in quantization literature for Transformer models, is insufficient to address the outlier issue. The authors design a Karhunen-Loeve Transformation enhanced rotation to control the variance for offline weight quantization and Smooth-Fused rotation to control the variance for online activation quantization.

**Strengths:**

1. The KLT enhanced rotation for controlling the variance is interesting.
2. The experiments show significant improvement compared to quantization methods proposed for Transformer models.

**Weaknesses:**

See question section.

**Questions:**

1. I think it is possible (also easier) to control the variance of Hadamard transformed weight by simply applying a diagonal scaling matrix so that the variance of each row is the same. This is similar to per channel quantization, where different scalings are applying to different rows of weight matrix before doing rounding. I am wondering how does this scheme compare to the proposed method.
2. The visualized value distribution in Figure 1 looks very similar to the visualization for Transformer models. More discussion on why quantization for Transformer models failed for mamba would be interesting.
3. 8 bit quantization for Transformer models is quite easy without noticeable performance drop, but it looks like that quantization for mamba is hard based on the experiments. Can the authors comment on this?

---

> ### Author Response · Authors · 2024-11-22
> **Response to Reviewer KhMz Part1**
>
> We sincerely thank you for your valuable time and efforts in reviewing our manuscript, and we tried our best to address all your concerns as follows.
>
> > I think it is possible (also easier) to control the variance of Hadamard transformed weight by simply applying a diagonal scaling matrix. How does this scheme compare to the proposed method.
>
> Our KLT-enhanced rotation technique stands out by ensuring **both** consistent variances and uniform maximum values **concurrently**.
>
> (1) Constrains of the simple scheme.
>
> A diagonal scaling matrix does help in aligning variances. However, neither applying it before nor after the Hadamard rotation can simultaneously ensure consistent variances and uniform maximum values. Given input data $X$, diagonal scaling matrix $S$ with variances as values, and the Hadamard matrix $H$, we can establish three transformed rotation matrices: $SH$, $HS$, and $KH$ (ours).
> 1. For $SH$ rotation, through multiplying $X$ with $S$, the variances can be temporarily aligned. However, this occurrence will be immediately disturbed by multiplying $H$, as Hadamard rotation can **change the values of variances**.
> 2. For $HS$ rotation, through multiplying $X$ with $H$, the maximum values can be temporarily uniformed. However, this occurrence will be immediately disturbed by multiplying $S$, as the value difference of $S$ can **lead into new outliers**.
>
> (2) Effectiveness of our approach.
>
> We propose KLT-enhanced rotation, which overcome the above-mentioned constrains. Considering the Equation (9) in the manuscript,
> $$H_K=KH,$$
> and the rotation to data $X$ can be described as:
> $$XH_K=XKH.$$
> Through multiplying $X$ with $K$, the distribution of $X$ is transformed but **not variances-aligned yet**. By multiplying $H$, the $XK$ is endowed with uniformed maximum values (owing to the property of Hadamard transformation [1]). **At the same time**, due to the transformed distribution of $X$,  the final channel variances are also aligned (as detailedly proved in Section 4.2 of the manuscript).
>
> (3) Further proof via experimental results.
>
> Following the experimental settings in Section 5 of the manuscript, we quantize the Vision Mamba models [2] in W8A8 setting by using different rotation matrices. Results show that our KLT-enhanced rotation (KH) outperforms other simple schemes.
> | Rotation Matrix | Average      | Vim-T    | Vim-T$^\dagger$ | Vim-S    | Vim-S$^\dagger$ |
> | :---------------: | :------------: | :--------: | :----------------: | :--------: | :----------------: |
> | $SH$         | 78.0         | 74.7     | 76.6             | 79.6     | 81.0             |
> | $HS$           | 77.9         | 74.1     | 76.1             | 80.2     | 81.2             |
> | $KH$           | **78.8** | **75.6** | **77.8**         | **80.3** | **81.4**         |

---

> > ### Author Response · Authors · 2024-11-22
> > **Response to Reviewer KhMz Part2**
> >
> > > Why do quantization methods for Transformers fail for Mamba despite the similar visualized distribution?
> >
> > Mamba models typically have a more uneven distribution than Transformers for the following reasons.
> >
> > **(1) Distribution difference increases by the block index.**
> >
> > In Figure 1 of the manuscript, we exhibit the most common cases of outliers, e.g., the former blocks. However, we observe that the distribution gap between Mamba and Transformers becomes more and more significant as the block index increases. For instance, we randomly sample 96 images from ImageNet, then feed them into the classic Vit model [3] and the Vision Mamba (Vim) model. Next, we extract the top-3 channel maximums and top-3 channel variances of the input activations of all quantized modules **in the last block**. We have added this part to the Appendix A.8.
> > | Model                 | Module                                      | Top3 Channel Maximums                        | Top3 Channel Variances                  |
> > | :---------------------: | :-------------------------------------------: | :--------------------------------------------: | :---------------------------------------: |
> > | vit-base-patch16-224  | attention.qkv_proj                          | 4.7 / 3.8 / 3.7                              | 0.4 / 0.4 / 0.4                         |
> > |                       | attention.qk_matmul.q | 7.1 / 7.1 / 7.1                             | 2.1 / 1.6 / 1.6                              |                                         |
> > |                       | attention.qk_matmul.k | 10.8 / 10.6 / 10.5                          | 4.5 / 4.4 / 4.3                              |                                         |
> > |                       | attention.o_proj      | 11.6 /  8.8 /  8.6                          | 4.1 / 3.4 / 3.3                              |                                         |
> > |                       | attention.pv_matmul.p | 4.8 / 4.7 / 4.2                             | 0.1 / 0.1 / 0.1                              |                                         |
> > |                       | attention.pv_matmul.v | 14.9 / 12.5 / 12.5                          | 6.5 / 5.4 / 4.5                              |                                         |
> > |                       | mlp.fc1               | 11.4 / 8.2 /  7.8                           | 2.3 / 2.0 / 1.6                              |                                         |
> > |                       | mlp.fc2               | 17.26 / 13.15 / 13.01                       | 12.10 /  9.84 /  7.35                        |                                         |
> > | vim-base-patch16-224  | in_proj                                     | **49.3** **/48.8 /** **38.9**                | **230.8** **/** **95.5** **/** **62.8** |
> > |                       | conv1d                | **48.2** **/** **36.2** **/** **36.2**      | **41.3** **/** **27.7** **/** **27.8**       |                                         |
> > |                       | x_proj                | 15.1 / 12.0 / 11.1                          | 8.6 / 4.5 / 3.9                              |                                         |
> > |                       | dt_proj               | 14.9 / 13.6 / 4.1                           | 11.0 / 10.9 / 10.6                           |                                         |
> > |                       | matmul_in1            | **75.2** **/** **59.0** **/** **58.2**      | **20.8** **/** **19.84** **/** **18.4**      |                                         |
> > |                       | matmul_in2            | 8.5 / 6.7 / 5.4                             | 1.8 /1.6 / 1.6                               |                                         |
> > |                       | out_proj              | **1371.6** **/** **1064.3** **/** **930.3** | **8854.9** **/** **2513.8** **/** **1377.8** |                                         |
> >
> > (2) Negative effect of the PScan.
> >
> > As stated in line 70-76 of the manuscript, the PScan operator, which is unique for the SSM-based Mamba structure, expands the data distribution range of activations. This behavior can increases the channel variances, consequently revealing the shortcomings of the Hadamard rotation. We have added several **visualized cases in Appendix A.9 of the manuscript**.
> >
> > (3) Less normalization.
> >
> > Mamba structures usually contain less normalization layer than Transformers. For instance, in the qwen [4] and llama [5] models, there are two normalization layers in each transformer block, but there is **only one normalization layer** in the mamba block. As the normalization operation is able to uniform the activations, less application means less regular distribution and more quantization difficulties.

---

> > > ### Author Response · Authors · 2024-11-22
> > > **Response to Reviewer KhMz Part3**
> > >
> > > >Why 8-bit quantization is easy for Transformers but hard for Mamba?
> > >
> > > As described in the second response, Mamba networks generate larger range of activations, especially with the usage of the PScan operator. Also, they have less normalization layers to uniform the data distribution. These features cause great challenges for quantization even at 8-bit precision.
> > >
> > > At present, there is almost no quantization research on the Mamba architecture that can achieve a satisfying performance, which gives rise to our motivation and contribution.
> > >
> > > ---
> > > If there are still any unresolved doubts, please feel free to let us know, and we will make every effort to solve them.
> > >
> > > **References**:
> > >
> > > [1] QuaRot: Outlier-Free 4-Bit Inference in Rotated LLMs.
> > >
> > > [2] Vision Mamba: Efficient Visual Representation Learning with Bidirectional State Space Model.
> > >
> > > [3] An Image is Worth 16x16 Words: Transformers for Image Recognition at Scale.
> > >
> > > [4] Qwen Technical Report.
> > >
> > > [5] LLaMA: Open and Efficient Foundation Language Models.

---

> > > > ### Author Response · Authors · 2024-12-02
> > > >
> > > > Dear Reviewer KhMz,
> > > >
> > > > We hope this message finds you well. We would like to sincerely thank you for your thoughtful review and valuable feedback on our paper. Your positive assessment and constructive comments have been instrumental in helping us improve our work.
> > > >
> > > > If you find that our revisions have satisfactorily addressed your concerns, we would be grateful if you could consider reflecting this in your final assessment.
> > > >
> > > > Thank you again for your valuable contributions to our research.
> > > >
> > > > Warm regards,
> > > >
> > > > The Authors

---

### Author Response · Authors · 2024-11-22
**General Response**

We sincerely thank all reviewers for your valuable time and insightful comments. The manuscript has been moderately revised in accordance with the suggestions received. Please feel encouraged to discuss with us if there are any unresolved doubts. Here is a list of updates in the order of pages, where added sentences or sections are marked in blue:
- Add futher explanation of the KLT's input (Section 4.2, concerned by Reviewer 5nSC).
- Add generalization evaluation of the KLT-enhanced rotation (Appendix A.7, suggested by Reviewer 5nSC and kvr1).
- Add experimental data to differentiate the activation distribution of Mamba and Transformers (Appendix A.8, concerned by Reviewer KhMz).
- Add visualized data to further prove the effect of the PScan (Appendix A.9, concerned by Reviewer KhMz).

---

### Meta-Review · Area_Chair_uM6F · 2024-12-19

**Metareview:**

The authors of this paper consider the problem of model quantization for Mamba family state-space models.  In particular, the authors observe that standard quantization approaches for CNNs or Transformers often perform poorly on Mamba models due to the presence of large outliers that can occur at various intermediate points within the model.  While one reviewer notes that this issue of outliers has been noted in previous works (posted to arxiv briefly before the ICLR deadline), the authors go beyond identifying the presence of outliers and study how the Mamba architecture can amplify these outliers and how to minimize this effect with their proposed quantification scheme.

The most negative review mentions questions about how steps of the method are calculated, but the description of this appears clear to me in the paper and the authors' response.  Likewise, other reviews did not identify any issues that I found to be particularly concerning, and overall I believe the paper will be of interest to the community.

**Additional Comments On Reviewer Discussion:**

One reviewer asked about the potential to extend this method to models beyond Mamba, for which the authors note that indeed it is a general method which can be applied to other models, but which is particularly beneficial for Mamba models due to the prevalence of outliers that are created in Mamba models.  The further demonstrate this with experiments showing strong quantification performance for a transformer model, which the reviewer found convincing.

---

### Decision · Program_Chairs · 2025-01-22

Accept (Poster)